# A strong bimetal-support interaction in ethanol steam reforming

Hao Meng [1], Yusen Yang [1] ✉, Tianyao Shen[1], Wei Liu[1], Lei Wang[1], Pan Yin[1], Zhen Ren[1], Yiming Niu [2], Bingsen Zhang [2], Lirong Zheng[3], Hong Yan[1], Jian Zhang [1] ✉, Feng-Shou Xiao [1,4] ✉, Min Wei [1] ✉ & Xue Duan[1]

The metal-support interaction (MSI) in heterogeneous catalysts plays a crucial role in reforming reaction to produce renewable hydrogen, but conventional objects are limited to single metal and support. Herein, we report a type of RhNi/TiO$_2$ catalysts with tunable RhNi-TiO$_2$ strong bimetal-support interaction (SBMSI) derived from structure topological transformation of RhNiTi-layered double hydroxides (RhNiTi-LDHs) precursors. The resulting 0.5RhNi/TiO$_2$ catalyst (with 0.5 *wt*.% Rh) exhibits extraordinary catalytic performance toward ethanol steam reforming (ESR) reaction with a H$_2$ yield of 61.7%, a H$_2$ production rate of 12.2 L h$^{-1}$ g$_{cat}^{-1}$ and a high operational stability (300 h), which is preponderant to the state-of-the-art catalysts. By virtue of synergistic catalysis of multifunctional interface structure (Rh-Ni$^{\delta-}$-O$_v$-Ti$^{3+}$; O$_v$ denotes oxygen vacancy), the generation of formate intermediate (the rate-determining step in ESR reaction) from steam reforming of CO and CH$_x$ is significantly promoted on 0.5RhNi/TiO$_2$ catalyst, accounting for its ultra-high H$_2$ production.

Supported metal catalysts with metal-support interaction (MSI) have been widely demonstrated in heterogeneous catalysis for many industrially important reactions (e.g., hydrogenation, oxidation, and catalytic reforming)[1–3]. Especially, the strong metal-support interaction (SMSI), a term to describe the phenomenon that the geometric and electronic structure of metal species is modified through interaction with supports has attracted considerable research interest for decades. More recently, Bao, Zhang, de Jong, and Christopher et al. reported a number of new strategies to regulate SMSI including redox-regulated (Pt/TiO$_2$, Co/ Nb$_2$O$_5$)[4,5], particle size-controlled (Ir/CeO$_2$, Ru/TiO$_2$)[6,7], adsorbate-mediated (Ni/TiO$_2$, Ru@MoO$_{3-x}$, and Rh/Nb$_2$O$_5$)[8,9], soft/wet chemistry-assisted (Au@TiO$_2$)[10,11] and crystal phase/facet-guided catalysts (Ru/TiO$_2$ and Pd/TiO$_2$)[12,13]. In these cases, effective supports with engineered characteristics have been successfully employed to stabilize metal species, modify its geometric/electronic

structure, and regulate mass transfer efficiency. Importantly, the proximity of active metal species to support defects also makes a great contribution to promote the activation adsorption of reactants, optimize the transition state of adsorbates and facilitate the transformation of reaction intermediates[14–16]. In general, the SMSI involved in heterogeneous metal catalysts plays a critical role in boosting catalytic performance toward structure-sensitive reactions.

Hydrogen (H$_2$) with a high energy density and zero pollution is being contemplated as the most promising alternative energy, which has been globally explored based on steam reforming of hydrocarbons and oxygenates (e.g., CH$_4$, CO, CH$_3$OH, and CH$_3$CH$_2$OH) in the preceding decades[3,17–29]. Ethanol, with low toxicity and high hydrogen/carbon ratio, which can be facilely produced from renewable biomass, has been employed as a hydrogen feedstock through ethanol steam reforming (ESR) reaction[18,24–29]. In respect to such a structure-sensitive

[1]State Key Laboratory of Chemical Resource Engineering, Beijing Advanced Innovation Center for Soft Matter Science and Engineering, Beijing University of Chemical Technology, Beijing 100029, P. R. China. [2]Shenyang National Laboratory for Materials Science, Institute of Metal Research, Chinese Academy of Sciences, Shenyang 110016, P. R. China. [3]Institute of High Energy Physics, Chinese Academy of Sciences, Beijing 100049, P. R. China. [4]Key Lab of Biomass Chemical Engineering of Ministry of Education, College of Chemical and Biological Engineering, Zhejiang University, Hangzhou 310027, P. R. China. ✉e-mail: yangyusen@buct.edu.cn; jianzhangbuct@buct.edu.cn; fsxiao@zju.edu.cn; weimin@mail.buct.edu.cn

reaction, the metal active sites are geometrically and electronically modified *via* SMSI, and the supports in most cases participate in the catalytic reaction[2,15]. Nevertheless, the conventionally reported SMSI is normally focused on individual metal and oxide support, whose single-channel interaction shows restriction when applied in certain complex processes[4,16], such as ESR reaction involving multiple sequential steps. Therefore, how to develop sophisticated catalysts based on multi-channel SMSI with largely enhanced catalytic performance for ESR still remains a huge challenge. Meanwhile, an in-depth insight into interfacial active sites at the atomic level is crucial for understanding reaction mechanism and pathway.

In this work, we prepared a series of $x$RhNi/TiO$_2$ catalysts with tunable strong bimetal-support interactions (SBMSI) based on a topotactic transformation from RhNiTi-layered double hydroxides (RhNiTi-LDHs) precursors. The CO-DRIFT and STEM confirm the existence of a significantly reversible TiO$_2$ coating over RhNi bimetallic nanoparticle. The coordination and electronic structure between RhNi bimetal and TiO$_2$ support are facilely regulated *via* changing the doping content of Rh, which realizes a multi-transfer pathway of electrons and further optimizes the interfacial active sites (Rh-Ni$^{\delta-}$-O$_v$-Ti$^{3+}$). The resulting 0.5RhNi/TiO$_2$ (with 0.5 *wt.*% Rh) sample exhibits a prominent catalytic performance toward ESR reaction, with an ethanol conversion of 99.7% and a H$_2$ yield of 61.7% at 400 °C. The H$_2$ production rate reaches 12.2 L h$^{-1}$ g$_{cat}^{-1}$ with an outstanding catalysis stability within 300 h, which is preponderant to the state-of-the-art catalysts. The steam reforming of CO and CH$_x$, which is the crucial step affecting H$_2$ production in ESR, is largely boosted at the interfacial sites (Rh-Ni$^{\delta-}$-O$_v$-Ti$^{3+}$). Kinetic studies combined with in situ characterizations (XAFS and FT-IR) and DFT theoretical calculations demonstrate that the SBMSI in 0.5RhNi/TiO$_2$ catalyst not only promotes the formation of formate intermediate (the rate-determining step), but also alleviates the strong binding of formate and CO$_2$ on catalyst surface, both of which are beneficial to the transformation of reactants and the regeneration of active sites.

## Results and discussion
### Synthesis and characterization of $x$RhNi/TiO$_2$ catalysts
The XRD patterns show a series of typical characteristic diffraction peaks of hydrotalcite-like structure (003, 006, 012, and 110) in as-synthesized $x$RhNiTi-LDHs precursors (Supplementary Fig. 1a). As shown in the SEM images (Supplementary Fig. S2), the introduction of trace Rh does not significantly affect its surface topography, and all these samples display a frizzy flowerlike morphology. After a treatment in H$_2$ atmosphere at 400 °C, the crystallite surface becomes rough with enhanced specific surface area and enriched pore structure (Supplementary Fig. 3 and Supplementary Table 1). The crystal structure of as-obtained $x$RhNi/TiO$_2$ samples is indexed to a super-imposition of face-centered cubic (fcc) Ni (JCPDS 7440-02-0) and anatase TiO$_2$ phase (JCPDS 21-1272) (Supplementary Fig. 1b). Transmission electron microscopy (TEM) and high-resolution transmission electron microscope (HR-TEM) images of $x$RhNi/TiO$_2$ samples show that Ni nanoparticles (particle size: 12–16 nm) are uniformly dispersed within the TiO$_{2-x}$ matrix (Supplementary Figs. 4 and 5). Especially, a well-defined metal-support (Ni-TiO$_2$) interface structure is clearly observed in these $x$RhNi/TiO$_2$ samples (Supplementary Fig. 5 and Supplementary Note 1).

### Catalytic performance and kinetic analysis toward ESR
The ESR reaction was performed in a fixed-bed reactor with steam/carbon (S/C) ratio of 3. Compared with Ni/TiO$_2$ and Rh/TiO$_2$, the RhNi/TiO$_2$ and Rh/Ni catalysts remarkably promote the conversion of ethanol and acetaldehyde at 350 °C (Supplementary Figs. 6a, b), indicating the advantages of RhNi bimetal system. As the reaction temperature reaches 400 °C, an almost complete conversion of ethanol and acetaldehyde is obtained for all these catalysts, but the distribution of gas product (CO, CO$_2$, CH$_4$, and H$_2$) is significantly different (Fig. 1b and

Supplementary Fig. 6c). This indicates that the cleavage of C−C, O−H and C−H bonds in ethanol and acetaldehyde is sensitive to reaction temperature, and 400 °C is required for the breakage of the chemical bonds above. For the $x$RhNi/TiO$_2$ catalysts, the H$_2$ yield shows a volcanic trend as Rh content increases from 0 to 1%, and the maximum value is present in 0.5RhNi/TiO$_2$ sample (H$_2$ yield: 61.7%; production rate: 12.2 L h$^{-1}$ g$^{-1}$) (Fig. 1c and Supplementary Fig. 7), which is preponderant to the state-of-the-art catalysts for ESR (Supplementary Table 2). Furthermore, we also tested the ESR reaction at a higher WHSV (21 h$^{-1}$) and GHSV (16700 h$^{-1}$) at 400 °C (ethanol conversion less than 70%), and the 0.5RhNi/TiO$_2$ catalyst still displayed the optimal hydrogen yield and relatively low CH$_4$ and CO yields (Supplementary Fig. 8). This is in sharp contrast to the Rh/NiO and Rh/TiO$_2$ samples, which give a much lower H$_2$ yield but a higher proportion of CO and CH$_4$ species (Fig. 1b), showing the superiority of bimetal-support (RhNi-TiO$_2$) system. In addition, the time on stream (TOS) tests for Ni/TiO$_2$, Rh/Ni, Rh/TiO$_2$, and 0.5RhNi/TiO$_2$ were carried out at 400 °C, respectively. After the reaction for 40 h, the ethanol conversion over Ni/TiO$_2$, Rh/Ni, and Rh/TiO$_2$ decreases from 100% to 91.33%, 58.54%, and 74.16%, respectively (Supplementary Fig. 9). In contrast, both the ethanol conversion and hydrogen yield in the presence of 0.5RhNi/TiO$_2$ catalyst remain stable within 300 h (Fig. 1d). The crystal structure and particle size of the used catalyst do not show obvious change except for the formation of certain carbon-accumulating species (Supplementary Fig. 10).

The *operando* infrared spectra (Supplementary Figs. 11–13 and Supplementary Note 2) were performed to identify the ESR reaction path. The results from the infrared spectra verify that the ESR reaction over 0.5RhNi/TiO$_2$, Ni/TiO$_2$, and Rh/Ni catalysts undergoes ethanol dehydrogenation to acetaldehyde followed by acetaldehyde decomposition to CO and CH$_x$ (CO/CH$_x$-mediated reforming process), rather than the acetate paths (Fig. 1a)[24,28,29]. The resulting CO and CH$_x$ as key intermediates continue to react with H$_2$O to produce CO$_2$ and H$_2$. This is consistent with the results from the *operando* pulse experiment of ethanol and water (Fig. 1h–j) with mass spectrometer (MS), where the characteristic bands assigned to various reaction intermediates are observed whilst acetate species are not detected during the whole reaction process. Thus, the H$_2$ yield depends heavily on the transformation of these CO and CH$_x$ intermediates, in addition to ethanol conversion.

Furthermore, we performed kinetic tests on ethanol dehydrogenation, acetaldehyde decomposition, steam reforming of CO (or CH$_4$) to study the C−H bond cleavage, C−C bond cleavage, CO or CH$_x$ transformation during ESR reaction. As shown in Supplementary Fig. 14a, the apparent activation energy of these reaction processes gives the following sequence: ethanol dehydrogenation (44.54 kJ mol$^{-1}$) < acetaldehyde decomposition (50.45 kJ mol$^{-1}$) < CO steam reforming (89.46 kJ mol$^{-1}$) < CH$_4$ steam reforming (101.26 kJ mol$^{-1}$), which indicates that the cleavage of C−H and C−C bonds in ethanol is facile whilst the transformation of intermediates (CO and CH$_x$) is rather difficult. This result is further demonstrated through a more significantly concentration-dependent reaction order for CO and CH$_4$ in comparison with ethanol and acetaldehyde: ethanol (0.46) < acetaldehyde (0.62) < CO (1.18) < CH$_4$ (1.23) (Supplementary Fig. 14b). In addition, we contrasted the conversion rates of CO and CH$_4$ over these samples (Fig. 1e). The CO conversion declines in the following sequence: Rh/Ni > 0.5RhNiTi > NiTi >> Rh/TiO$_2$; whilst CH$_4$ conversion decreases in order: 0.5RhNiTi > NiTi > Rh/Ni > Rh/TiO$_2$. The higher CO conversion rate on Rh/Ni is not consistent with its high CO yield. Considering that the reaction mixture would reach equilibrium at prolonged reaction time, and the products (H$_2$ and CO$_2$) may affect the progress of main reaction, we measured the reaction rate of water-gas shift (WGS) reaction by adding a small amount of H$_2$ or CO$_2$ into the reaction atmosphere (Fig. 1f). A similar decrease extent is observed for these catalysts after H$_2$ introduction (Fig. 1g), indicating that the

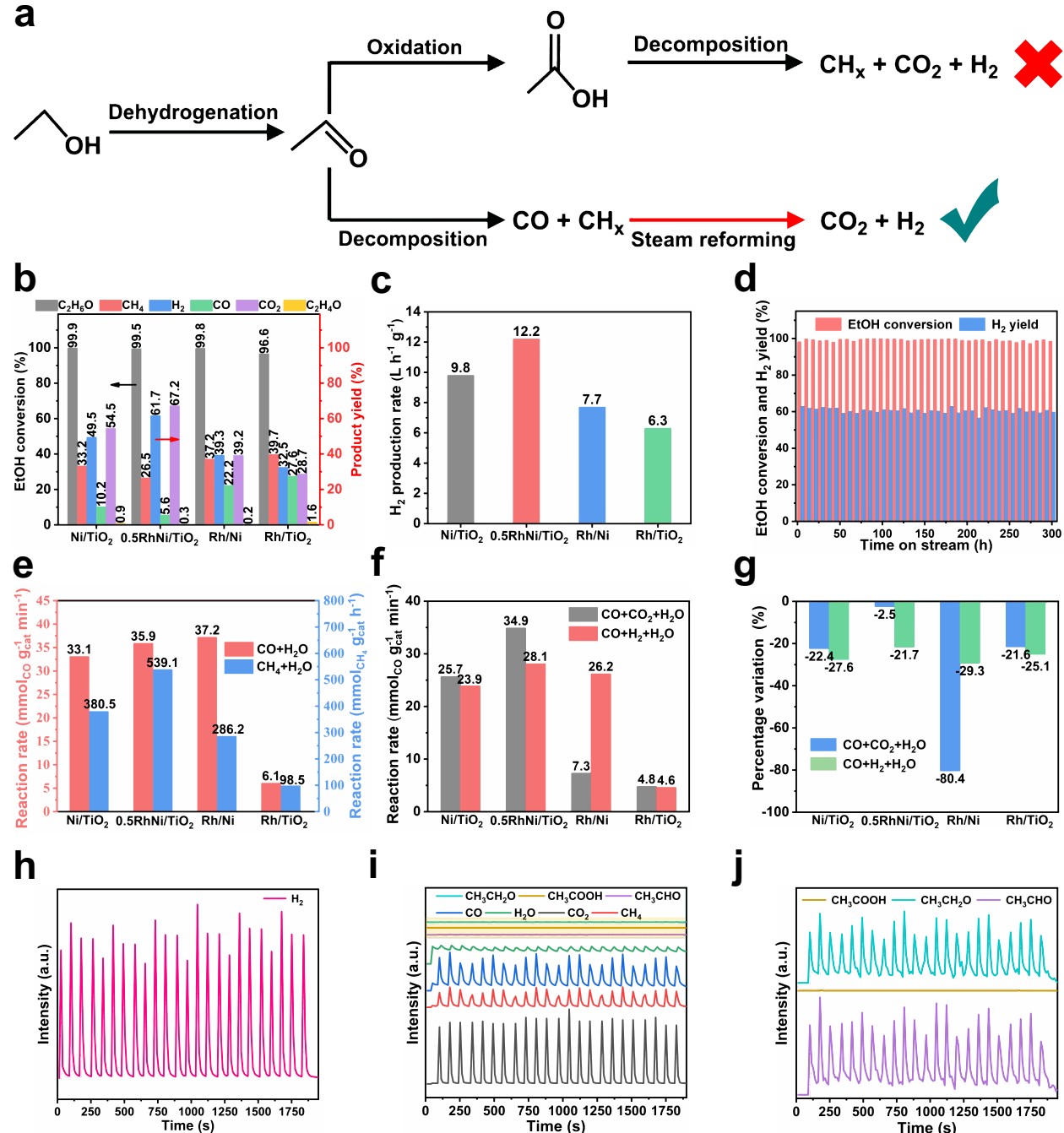

**Fig. 1 | Catalytic performance toward ESR reaction and kinetic studies on CO and CH₄ reforming. a** Schematic representation for the reaction paths of ESR reaction (Red arrow indicates the key step). **b** Ethanol conversion, products yield and **c** H₂ production rate over various catalysts (reaction conditions: catalyst (0.15 g) + SiO₂ (1.50 g); liquid feed of S/C = 3 at 0.060 mL min⁻¹; N₂ carrier at 50.0 mL min⁻¹; reaction temperature: 400 °C; time on stream: 1.5 h). **d** Time on stream (TOS) test of 0.5RhNi/TiO₂ catalyst at 400 °C. **e, f** Reaction rate for CO/CH₄ steam reforming reaction within kinetic range (CO or CH₄ conversion <10%).

Reaction conditions: catalyst (5 mg) + SiO₂ (50 mg), liquid feed of H₂O at 0.032 mL min⁻¹, CO at 50.0 mL min⁻¹, CH₄ at 10 mL min⁻¹, CO₂/H₂ at 10.0 mL min⁻¹, N₂ carrier at 50 mL min⁻¹, reaction temperature: 400 °C, time on stream: 0.5 h. **g** Percentage variation in reaction rate of CO steam reforming with CO₂/H₂ addition over Ni/TiO₂, 0.5RhNi/TiO₂, Rh/Ni, and Rh/TiO₂ catalysts, respectively. **h, i** MS signals from the *operando* pulse experiment of ethanol and water (S/C = 3) over 0.5RhNi/TiO₂ catalyst at 400 °C. **j** Enlarged view for MS signals of the three species in **i**.

inhibitory effect mainly originates from the equilibrium limit (Le Chatelier's principle). In contrast, these catalysts show a significantly difference in CO reaction rate after CO₂ introduction: a remarkable decline ratio of 80.4% (from 37.2 to 7.3 mol CO g$_{cat}$⁻¹ min⁻¹) is obtained over Rh/Ni catalyst, relative to 0.5RhNi/TiO₂ with a slight decrease of 2.5%, indicating the existence of other important factors besides equilibrium limit. Moreover, the CO₂-TPD (Supplementary Fig. 15 and Supplementary Note 3) and reaction order measurements

(Supplementary Fig. 16 and Supplementary Note 4) confirm that Rh/Ni catalyst shows a stronger CO₂ adsorption ability and a more negative CO₂ reaction order (−0.64) compared with 0.5RhNi/TiO₂ (−0.18), demonstrating a stronger inhibitory action by CO₂ in the former case. In addition, the conversion of CO and CH₄ are also determined when H₂O, CO, CH₄, H₂, and CO₂ are present in the reactants simultaneously (Supplementary Fig. 17 and Supplementary Note 5), and 0.5RhNi/TiO₂ catalyst still exhibits the highest conversion of CH₄ and CO. Thus, the

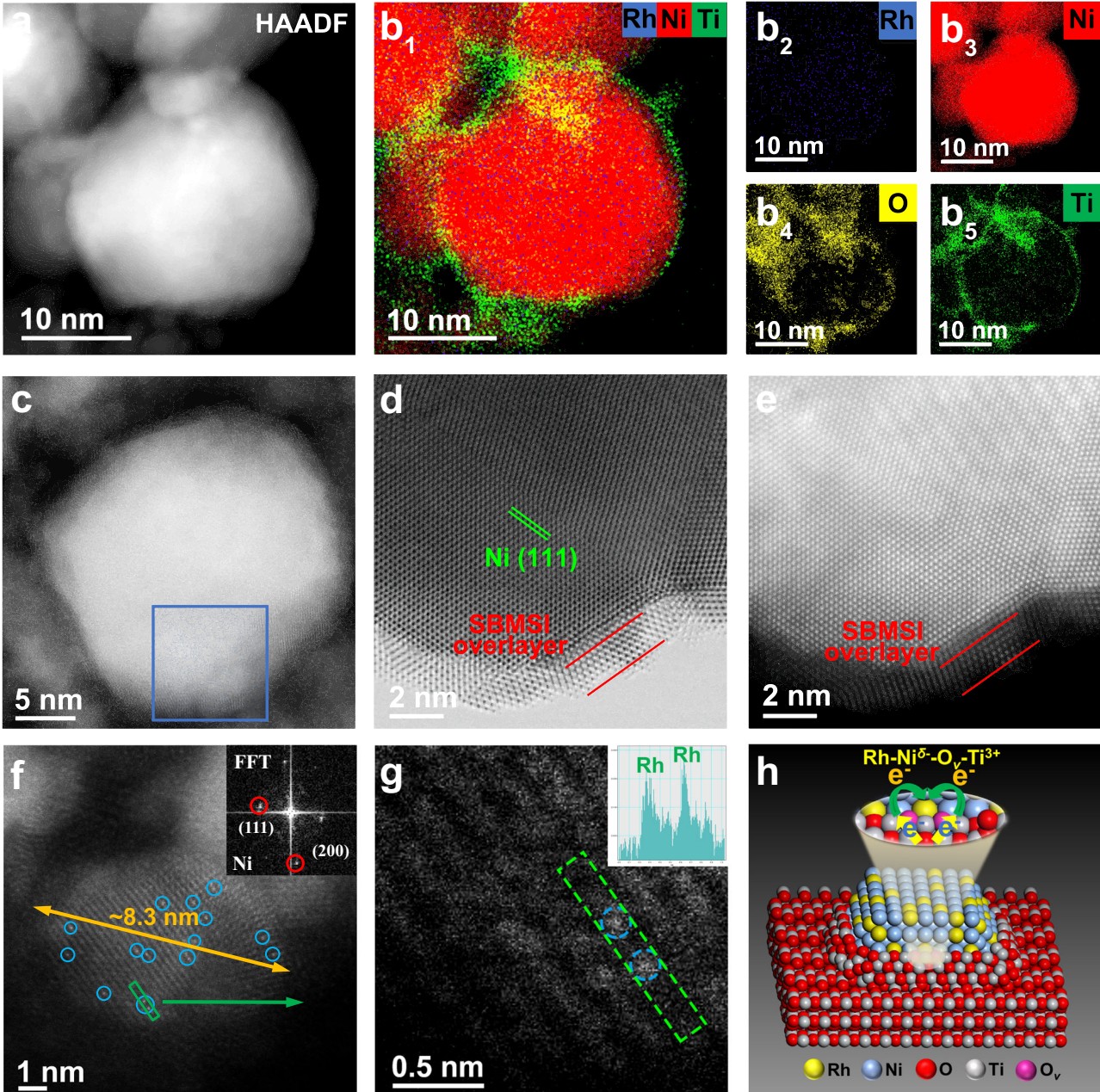

**Fig. 2 | Microstructure fine-structure characterizations. a, c, d** ac-HAADF-STEM and **e** BF-STEM images, **b₁–b₅** corresponding EDS mapping, and **f, g** high-resolution STEM images of the 0.5RhNi/TiO₂ catalyst with atomically dispersed Rh species marked by blue circles. **h** Schematic diagram of the 0.5RhNi/TiO₂ catalyst.

kinetic studies above substantiate that 0.5RhNi/TiO₂ catalyst possesses a powerful catalytic transformation ability toward CO and $CH_x$ as well as a strong resistance against $CO_2$, resulting in its extraordinary catalytic performance toward ESR.

## Microstructure investigations and fine-structure characterizations

Aberration-corrected high-angle annular dark-field scanning transmission electron microscopy (ac-HAADF-STEM) image and energy-dispersive spectroscopy (EDS) measurements were carried out to explore the interfacial structure of $x$RhNi/TiO₂ samples (Fig. 2 and Supplementary Figs. 18–23). For 0.3RhNi/TiO₂ and 0.5RhNi/TiO₂ catalysts, Rh tends to present an atomic-level dispersion on the surface of Ni particles to form RhNi bimetallic structure (Fig. 2f, g, and Supplementary Fig. 18), and a significant interface encapsulation of TiO₂ on the surrounding of RhNi nanoparticle is observed after H₂ activation

(Fig. 2b₁–b₅, d, e, and Supplementary Fig. 20). However, visible $TiO_{2-x}$ overlayer recedes over RuNi bimetallic NPs when exposed to air oxidation conditions (Supplementary Fig. 21), which is consistent with the phenomenon from classical SMSI[4,9,13,16,30–32]. For 0.8RhNi/TiO₂ and 1.0RhNi/TiO₂ sample with a higher Rh content, Rh species shows an aggregation state on the surface of Ni particles (Supplementary Figs. 22 and 23). The results intuitively confirm the well-defined interface structure between RhNi bimetal and TiO₂ support. In addition, Ni/TiO₂ and Rh/TiO₂ samples also show classic SMSI effect as demonstrated by STEM and elemental line scanning, in which the reversible overlayer of $TiO_{2-x}$ appears and disappears under the reduction and oxidation conditions, respectively (Supplementary Figs. 24–26).

The CO-DRIFT was used to further identify the structure evolution as demonstrated above (Fig. 3a–c, and Supplementary Fig. 27). The peaks at 2115, 2104, and 2096 cm⁻¹ are assigned to the CO adsorbed at Ni and Rh sites, and the bands at 2080 and 2046 cm⁻¹ are attributed to

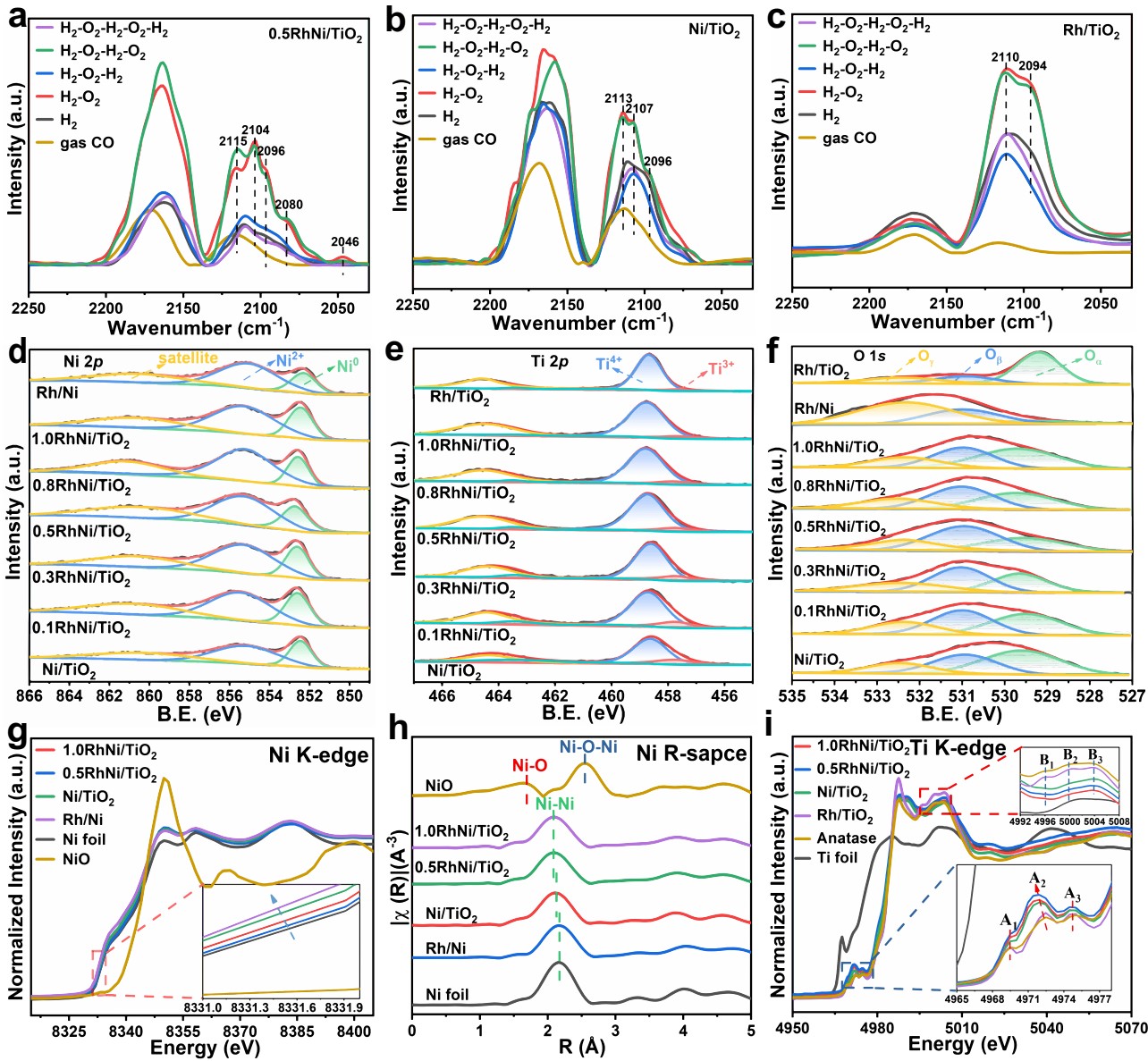

**Fig. 3 | Geometric and electronic properties characterizations.** CO-DRIFT spectra at ambiance temperature over **a** 0.5RhNi/TiO$_2$, **b** Ni/TiO$_2$, and **c** Rh/TiO$_2$ catalysts during five-cycle H$_2$ reduction-O$_2$ oxidation. XPS spectra of **d** Ni 2$p$, **e** Ti 2$p$, and **f** O 1$s$ for various samples. Normalized spectra of **g** Ni K-edge XANES, **h** Ni K-edge EXAFS at R-space, and **i** Ti K-edge XANES for various samples.

RhNi bimetallic interface sites. The alternative changes of weakened and strengthened CO adsorption intensity during five-cycle of H$_2$ reduction-O$_2$ oxidation correspond to a reversible encapsulation and decapsulation of TiO$_{2-x}$ overlayer. Notably, a more pronounced variation is observed over 0.5RhNi/TiO$_2$ catalyst, signifying a stronger interaction from RhNi bimetal and TiO$_2$ support. For Rh/Ni sample, the Rh clusters are dispersed on Ni without SMSI effect (Supplementary Figs. 27 and 28).

The electronic structure of $x$RhNi/TiO$_2$ samples was investigated by X-ray photoelectron spectroscopy (XPS). As the Rh content increases from 0 to 0.5%, the binding energy of Ni 2$p$ moves to higher energy, indicating a decreased Ni electron density on 0.5RhNi/TiO$_2$ catalyst (Fig. 3d and Supplementary Note 6). However, with a further increase of Rh loading from 0.5% to 1.0%, the binding energy of Ni shifts back to lower energy. The binding energy of Rh exhibits an opposite variation relative to Ni (Supplementary Fig. 29). The results indicate that the electron transfer from Ni to Rh becomes weak when the Rh distribution converts from atomic level to clusters or particles.

This volcanic change trend of electron density for Ni species verifies that this bimetal-support interface breaks through the traditional electron transfer mode between single metal and oxide support. Compared with Ni/TiO$_2$, the Ti$^{3+}$/(Ti$^{3+}$+Ti$^{4+}$) ratio in $x$RhNi/TiO$_2$ samples declines gradually with the increment of Rh loading (Fig. 3e). Notably, the Ti$^{3+}$ species in Rh/TiO$_2$ displays a significant reduction compared with Ni/TiO$_2$, indicating Ti$^{3+}$ mainly originates from Ni-TiO$_{2-x}$ interaction and the formation of RhNi bimetal would change such interaction. In addition, the O$_\beta$/(O$_\alpha$ + O$_\beta$ + O$_\gamma$) ratio (O$_\alpha$: lattice oxygen O$^{2-}$; O$_\beta$: chemisorbed oxygen O$_2^-$ or O$^-$; O$_\gamma$: other oxygen species including adsorbed water) shows a rise firstly and then a decline (Fig. 3f), and the maximal value is present in 0.5RhNi/TiO$_2$ (Supplementary Table 3), indicating that a lower Ni electron density promotes the combination of oxygen species. The variation in RhNi-TiO$_2$ interaction was further proved by H$_2$-TPR and H$_2$-TPD analysis (Supplementary Fig. 30 and Supplementary Note 7). The results above demonstrate that the electronic effect in $x$RhNi/TiO$_2$ samples can be finely regulated through tuning Rh content.

In addition, the interface electronic and coordination structure were studied *via* X-ray absorption spectroscopy (XAS). As shown in normalized XANES spectra of Ni K-edge (Fig. 3g), the absorption edge of all samples is located at lower photon energy relative to Ni foil, indicating the existence of a negative valence state ($Ni^{\delta-}$ species), which is similar to other $TiO_2$-supported group VIII metals with electron transfer from $TiO_{2-x}$ to the interfacial metal atoms[4,9,16,30–32]. Notably, the absorption edge of $0.5RhNi/TiO_2$ displays a minimum shift to lower energy among these samples, indicative of the lowest electron density of interfacial Ni atom. Meanwhile, the fitting results from FT $k^3$-weighted Fourier transforms of the extended X-ray absorption fine-structure (EXAFS) spectra and wavelet transforms (Fig. 3h, Supplementary Figs. 31 and 32, and Supplementary Table 4) show that the Ni–Ni bond length in $Ni/TiO_2$ sample becomes shorter compared with Ni foil and Rh/Ni, owing to the interaction between Ni and $TiO_2$ support. Moreover, the Ni–Ni bond length in $0.5RhNi/TiO_2$ and $1.0RhNi/TiO_2$ shortens further relative to $Ni/TiO_2$.

The normalized Ti K-edge XANES data are presented in Fig. 3i. All the samples show similar curves to the anatase reference in pre-edge range (4968 to 4980 eV) and post-edge region (4992 to 5008 eV), but the $A_1$ peak and the three peaks (denoted as $B_1$, $B_2$ and $B_3$) for $Ni/TiO_2$, $0.5RhNi/TiO_2$ and $1.0RhNi/TiO_2$ samples are less resolved; and the intensity of their $A_2$ peaks becomes stronger accompanied with a shift to lower energy. The results indicate the presence of distorted octahedral Ti–O environment associated with oxygen vacancies[13,33]. Fourier transform of Ti K-edge EXAFS spectra display that the first shell of Ti–O bond in $xRhNi/TiO_2$ samples gives an obviously less than sixfold coordination number (Supplementary Fig. 33, Supplementary Table 5 and Supplementary Note 8), which is the most significant for $0.5RhNi/TiO_2$. Based on the results afore-mentioned, a unique geometric and electronic structure between RhNi bimetal and $TiO_2$ support in $xRhNi/TiO_2$ catalysts is demonstrated clearly (Fig. 2h), defined as the strong bimetallic-support interaction (SBMSI), which is responsible for the outstanding catalytic performance of $0.5RhNi/TiO_2$.

## In situ spectral characterization for reaction mechanism

Based on the catalytic evaluations and kinetic studies, the key steps determining $H_2$ generation is the transformation of intermediate products (CO and $CH_x$). A series of in situ DRIFT measurements were carried out to investigate the influence of SBMSI on CO and $CH_x$ transformation. As $0.5RhNi/TiO_2$ is exposed to CO at 400 °C, $CO_2$ adsorption bands within 2270–2389 $cm^{-1}$ appear[34,35], whose intensity enhances gradually along with time, indicating the occurrence of CO disproportionation on the catalyst surface to produce $CO_2$ and C species (Supplementary Fig. 34 and Supplementary Note 9). Subsequently, a switching to $H_2O$ pulse leads to the decline of CO signal, accompanied by the formation of $Ti^{3+}$–OH and $Ti^{4+}$–OH species (3733–3594 $cm^{-1}$); and several absorption bands including formate species (2953–2857, 1581, and 1380 $cm^{-1}$) and bidentate hydrogencarbonates species (1457 and 1272 $cm^{-1}$) are observed (Fig. 4a, b)[1,9,36,37]. The peak intensity of formate enhances firstly and then declines with the pulse of water, indicating that such species is an important reaction intermediate. To further identify the generation path for formate intermediate, another $H_2O$ pulse test is implemented again after purging the surface CO and $CO_2$ by He (Supplementary Fig. 35). The analogous formate intermediate is detected, indicating that formate is derived from the further transformation of carbon species produced by CO disproportionation. In situ Raman was further used to verify this issue (Fig. 4c), in which two remarkable bands at 1330 and 1609 $cm^{-1}$ ascribed to D and G bands of carbon species were observed after purging CO[18,38,39]. Afterwards, the band intensity of carbon species declines, and the Ti–O signal increases for $0.5RhNi/TiO_2$ catalyst with the introduction of saturated water vapor. Furthermore, when CO and saturated water vapor are introduced together, very few carbon species are detected, indicating that accumulation and consumption of

carbon maintain a balance under WGS reaction conditions (Supplementary Fig. 36).

Furthermore, the TPSR-mass spectral analysis is carried out. Firstly, the $0.5RhNi/TiO_2$ catalyst was pretreated in $H_2$ at 400 °C for 30 min followed by He flushing for 30 min. Then, the gas was switched to CO, He, and saturated water vapor in turn to monitor the gas signals (Fig. 4g–i). The formation of $CO_2$ was detected synchronously with the introduction of CO, indicating the occurrence of CO disproportionation to generate $CO_2$ and C species (Fig. 4h), in accordance with the results of in situ DRIFT (Fig. 4a and Supplementary Figs. 34 and 35a) and in situ Raman spectra (Fig. 4c). Subsequently, He flushing followed by a saturated water vapor pulse over the catalyst gave rise to the generation of $H_2$ and $CO_2$ (Fig. 4i). Such phenomenon demonstrates that $H_2O$ dissociation occurs on oxygen vacancy of $TiO_2$ to generate active hydroxyl/oxygen groups, which further react with surface carbon species to give formate intermediate; subsequently, formate species undergoes decomposition to produce $CO_2$ and $H_2$. The results above demonstrate that the WGS reaction obeys an associative mechanism on the surface of $0.5RhNi/TiO_2$.

For $Ni/TiO_2$ catalyst, a similar associative reaction mechanism was also verified (Supplementary Figs. 38a and 39), except a lower intensity of formate intermediate, corresponding to its lower CO conversion (Supplementary Note 11 and Supplementary Note 12). For Rh/Ni catalyst without SBMSI, CO disproportionation process occurs; however, no associative reaction intermediate is detected after purging $H_2O$ pulse as proved by in situ DRIFT and in situ Raman results (Supplementary Fig. 40 and Supplementary Note 13). Furthermore, the mass spectral analysis confirms that $H_2O$ experiences dissociation over Rh/Ni catalyst to produce $H_2$ and active O species, and then O species reacts with CO or C species to form $CO_2$ (Supplementary Fig. 37 and Supplementary Note 10), which accords with redox mechanism[20,40–42]. Moreover, in situ Raman spectra also confirm the existence of carbonate species (1000–1200 $cm^{-1}$) on Rh/Ni and $0.5RhNi/TiO_2$ catalyst. The more significant intensity in the former case originates from a stronger binding of $CO_2$ (Fig. 4c and Supplementary Fig. 40c), which is consistent with the test results. The impact of $CO_2$ was discussed in detail based on in situ DRIFT, in situ Raman and *Quasi*-in situ XPS in supporting information (Supplementary Figs. 41–44 and Supplementary Notes 14–16), where $0.5RhNi/TiO_2$ with SBMSI shows the strongest $CO_2$ resistance. For $Rh/TiO_2$ catalyst, CO disproportionation does not occur along with fewer reaction intermediates after $H_2O$ pulse (Supplementary Figs. 38c and 45, and Supplementary Note 17), which is associated with its poor WGS reaction activity.

In addition, we also carried out in situ DRIFT investigations for $CH_4$ steam reforming on these catalysts to imitate the transformation of $CH_x$ species. After purging $CH_4$, an obvious peak appears at 1542 $cm^{-1}$ assigned to bidentate formate intermediate on $0.5RhNi/TiO_2$, $Ni/TiO_2$ and Rh/Ni samples (Supplementary Fig. 46). When introducing saturated water vapor into the reaction cell at 400 °C, formate intermediate declines accompanied with the formation of $CO_2$. Both the consumption rate of formate intermediate and the band intensity of $\delta(CH_x)$ at 1542 and 1302 $cm^{-1}$ decrease in the following order: $0.5RhNi/TiO_2 > Ni/TiO_2 > Rh/Ni$ (Supplementary Fig. 47 and Supplementary Note 18), in accordance with the test results (Fig. 1b, e). However, no obvious intermediate is found in the case of $Rh/TiO_2$ due to its poor catalytic performance (Supplementary Fig. 46h). Furthermore, no obvious carbon species is observed in in situ Raman spectra for $0.5RhNi/TiO_2$ and Rh/Ni catalysts, ruling out the complete dehydrogenation of $CH_x$ in reforming processes (Supplementary Fig. 48 and Supplementary Notes 19), since the most stable CH fragment can react with hydroxy group or active oxygen to produce formate intermediate ($CH_x \rightarrow CH + H_{x-1} \rightarrow HCOO^-$)[37,43–45]. After switching to a saturated water vapor, carbonate species (1000–1200 $cm^{-1}$) resulting from $CO_2$ behaves a stronger binding onto Rh/Ni than $0.5RhNi/TiO_2$ catalyst

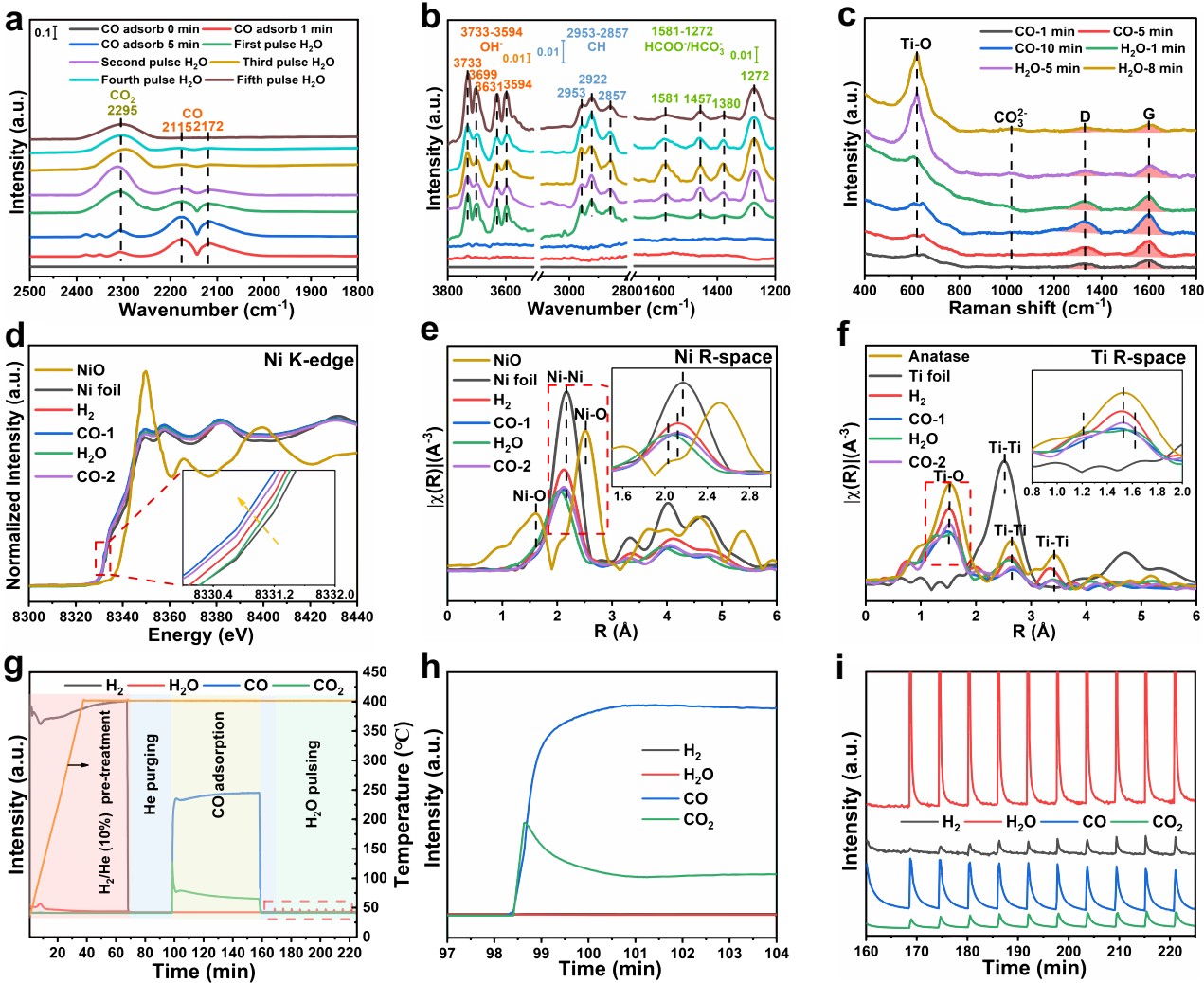

**Fig. 4 | In situ characterizations and reaction mechanism of CO reforming.**
**a**, **b** In situ DRIFT spectra, **c** in situ Raman spectra, **d** in situ normalized XANES spectra, **e** in situ Fourier-transform EXAFS spectra at Ni K-edge in R-space, and **f** Ti K-edge in R-space of CO adsorption and then $H_2O$ adsorption on 0.5RhNi/TiO₂ catalyst at 400 °C ($H_2$, CO-1, $H_2O$, and CO-2 denote the spectra after successive $H_2$ pretreatment, CO adsorption, $H_2O$ adsorption, and CO adsorption again for 20 min on 0.5RhNi/TiO₂ catalyst, respectively). **g** Mass spectral analysis for CO and $H_2O$ pulse over 0.5RhNi/TiO₂ catalyst and **h**, **i** corresponding local magnification regions in **g**.

(Supplementary Fig. 48), similar to the CO reforming process. Owing to the similar O−C−O structure of formate to $CO_2$ or carbonate, the lower conversion of $CH_x$ on Rh/Ni is possibly associated with the stronger binding of formate to Ni sites, which inhibits its further transformation. Therefore, the 0.5RhNi/TiO₂ catalyst with unique coordination and electronic structure resulting from SBMSI reduces the binding ability of species with analogous $COO^-$ structure ($CO_2$, carbonate, or formate), and thus promotes the generation and transformation of formate intermediate, accounting for its excellent reforming activity of CO and $CH_x$.

In situ XAFS measurements were performed to track the dynamic evolution of coordination structure and electronic state of bimetal-support interface sites during steam reforming process. For the 0.5RhNi/TiO₂ sample, when CO is introduced into the reaction system (CO-1), the absorption edge of Ni shifts to lower energy (Fig. 4d), accompanied with a decrease in the coordination number of Ni−Ni bond (Fig. 4e, Supplementary Fig. 49b and Supplementary Table 6). The corresponding variations in Ti K-edge XAFS spectra are also observed, where the A₂ peak shifts to lower energy (Supplementary Fig. 49c) with a decline in the coordination number of Ti−O bond (Fig. 4f). This is attributed to the catalyst reconstruction resulting from

CO disproportionation, in which the generated carbon species fixed to Ni sites reacts with active oxygen species offered by TiO₂ at interface sites. Once the reaction atmosphere is switched to a saturated water vapor ($H_2O$), the absorption edge of Ni moves toward higher energy and Ni−Ni bond length becomes shorter (Fig. 4d, e), due to the formation of formate intermediate on the catalyst surface. In addition, the intensity of A₂ peak in Ti K-edge XANES spectra (Supplementary Fig. 49c) declines and the Ti−O bond splits upon exposure to $H_2O$ vapor (Fig. 4f). This is ascribed to the quenching of oxygen vacancy in TiO₂ by active hydroxy species (from $Ti^{3+}-O_v$ to $Ti^{4+}-OH^-$), along with a structure reconfiguration. After purging CO again (CO-2), the catalyst structure is restored to its original state and a reaction cycle is completed. The results substantiate that the bimetal-support interfacial sites participate in the steam reforming processes, accompanied by the interface reconstruction in the case of 0.5RhNi/TiO₂ catalyst. In contrast, the metal sites on Rh/Ni catalyst play a dual role toward activation adsorption of both water molecule and reactants, corresponding to its poor catalytic properties. A detailed discussion on in situ XAFS measurements was offered in supplementary materials (Supplementary Fig. 50, Supplementary Table 7 and Supplementary Note 20).

Furthermore, we carried out first-principle calculations to further understand the roles of bimetal-support interfacial sites in ethanol conversion and steam reforming processes of CO and $CH_x$, and the $Rh_1Ni_7/TiO_{2-x}$ model systems was built (Supplementary Fig. 51 and Supplementary Note 21). Based on the Bader charge analysis results, compared with the $Ni_8/TiO_{2-x}$ (Supplementary Figs. 52a, c), the $Rh_1Ni_7/TiO_{2-x}$ system (Supplementary Figs. 52b, d) with SBMSI weakens the charge transfer from the oxygen vacancy on $TiO_{2-x}$ support to the $Rh_1Ni_7$ bimetallic interface (Supplementary Fig. 52e), which is consistent with the electronic structure characterization (Fig. 3d, g). For the ESR reaction mechanism, the reaction energy barriers for ethanol dehydrogenation and acetaldehyde decomposition were studied through DFT calculations. Based on the calculation results (Supplementary Figs. 53−62, Supplementary Table 8 and Supplementary Note 22), the optimal ethanol dehydrogenation path in $Ni_8/TiO_{2-x}$ and $Rh_1Ni_7/TiO_{2-x}$ systems follows $CH_3CH_2OH \rightarrow CH_3CH_2O^* \rightarrow CH_3CHO^* \rightarrow CH_3CO^*$, and then $CH_3CO^*$ undergoes C−C bond breaking to produce $CH_3^*$ and CO, which is consistent with the product distribution and *operando* characterizations (DRIFTS spectra and pulse experiment). Compared with $Ni_8/TiO_{2-x}$, the energy barriers for ethanol dehydrogenation and C−C bond cleavage decrease from 1.65 and 1.31 eV to 1.03 and 1.17 eV on $Rh_1Ni_7/TiO_{2-x}$, respectively. For the reaction mechanism of stream reforming of CO (Supplementary Figs. 63−72, Supplementary Table 8 and Supplementary Note 23), the potential energy profiles over $Rh_1Ni_7/TiO_{2-x}$ and $Ni_8/TiO_{2-x}$ catalysts are similar and shown in Fig. 5 and Supplementary Fig. 73, which mainly consists of four steps. Firstly, CO molecule undergoes activation adsorption at the hollow sites (adjacent to two Ni atoms and one Rh atom for $Rh_1Ni_7/TiO_{2-x}$; adjacent to three Ni atoms for $Ni_8/TiO_{2-x}$), and then dissociates to form C and O, followed by disproportionation reaction with another CO molecule to generate $CO_2$ and C species (blue dotted lines in Fig. 5 and Supplementary Fig. 73); subsequently, $H_2O$ molecule experiences activation adsorption at interfacial oxygen vacancy, which dissociates to active hydroxyl and oxygen species (green dotted lines in Fig. 5 and Supplementary Fig. 73); afterwards, the hydrogen from $H_2O$ dissociation binds to the carbon species to generate CH fragment, which is then attacked by active oxygen species to produce the $HCOO^-$ intermediate (red dotted lines in Fig. 5 and Supplementary Fig. 73); finally, formate undergoes decomposition to produce $CO_2$ and $H_2$ (orange lines in Fig. 5 and Supplementary Fig. 73). According to the calculation results, the formation of $HCOO^-$ intermediate is the rate-determining step with an energy barrier of 2.36 and 2.79 eV on $Rh_1Ni_7/TiO_{2-x}$ and $Ni_8/TiO_{2-x}$ catalysts, respectively. In addition, for the steam reforming processes of $CH_x$, the successive dehydrogenation of methyl occurs firstly on the surface of Ni and RhNi bimetal sites to generate CH fragment (Supplementary Figs. 74, 75 and Supplementary Note 24), followed by a similar process mentioned above for the transform of CH to $HCOO^-$ (Supplementary Figs. 67 and 71). In contrast, the formate formation from C/CH fragment shows an energy barrier of 2.36 and 2.79 eV on $Rh_1Ni_7/TiO_{2-x}$ and $Ni_8/TiO_{2-x}$ catalysts, respectively, much larger than that of ethanol dehydrogenation (1.03 and 1.65 eV) and acetaldehyde decomposition (1.17 and 1.31 eV), indicating that the transformation of CO and $CH_x$ is the crucial step, in accordance with the experimental results. Especially, the lower reaction energy barriers on $Rh_1Ni_7/TiO_{2-x}$ relative to $Ni_8/TiO_{2-x}$ verify that the ESR reaction is boosted at the bimetal-support interface sites, in well agreement with the catalytic evaluations.

In summary, we report a $RhNi/TiO_2$ catalytic system with well-defined SBMSI towards ESR reaction. The obtained $0.5RhNi/TiO_2$ catalyst gives exceptional hydrogen production ($H_2$ yield: 61.7%) and catalysis stability (300 h) at a relatively low temperature (400 °C). The microscopic fine-structure of $RhNi/TiO_2$ was studied by STEM, CO-DRIFT, and XAFS, in which the RhNi bimetallic nanoparticle with a reversible $TiO_2$ coating exhibited a multiple electron transfer pathway at the interfacial active sites ($Rh-Ni^{\delta-}-O_v-Ti^{3+}$). A comprehensive investigation including in situ spectroscopic characterizations, *operando* pulse experiments, kinetics studies, and DFT calculations substantiates that the ESR reaction in the presence of $0.5RhNi/TiO_2$ catalyst follows a $CO/CH_x$-mediated reforming process rather than the acetate path. This bimetal-support interface ($Rh-Ni^{\delta-}-O_v-Ti^{3+}$) plays a decisive role in the steam reforming of CO and $CH_x$ originating from ethanol dissociation, which is involved in the rate-determining step of ESR reaction. The modulated geometric and electronic structure of interfacial active sites resulting from SBMSI reduce the binding ability of species with $COO^-$ structure ($CO_2$, carbonate, or formate). This facilitates the generation and transformation of formate intermediate from $CO/CH_x$ reforming processes, which ensures a prominent hydrogen production rate and catalytic stability. The well-defined SBMSI demonstrated in this work can be extended to other structure-sensitive reactions involving multiple reaction substrates.

## Methods

### Chemicals and materials

Chemical reagents, including $Ni(NO_3)_2 \cdot 6H_2O$, $Al(NO_3)_3 \cdot 9H_2O$, tetra-butyl titanate, anatase, and urea were bought from Aladdin chemical reagent company. $RhCl_3 \cdot 3H_2O$ was purchased from Beijing HWRK CHEM; absolute ethanol was purchased from Tianjin DaMao chemical reagent factory. The above chemical reagents were used directly without further purification. Quartz sand ($SiO_2$, 40−60 mesh) was purchased from Tianjin Guangfu fine chemical research institute, and washed by using concentrated HCl prior to use. Deionized (DI) water with a resistivity of 18.2 MΩ cm was used in all experimental processes.

### Synthesis of catalysts

RhNiTi-LDHs (RhNiTi-layered double hydroxides) precursors were prepared *via* urea homogeneous precipitation method. Briefly, tetra-butyl titanate (0.013 mol $L^{-1}$), $Ni(NO_3)_2 \cdot 6H_2O$ (0.040 mol $L^{-1}$), and urea (0.500 mol $L^{-1}$) were dissolved in deionized water (150 mL) with vigorous stirring for 8 h at refluxing temperature (95 °C). After 40 min of reaction, a certain amount of $RhCl_3 \cdot 3H_2O$ aqueous solution (7.6 mg $mL^{-1}$) was slowly dripped into above solution. The resulting precipitate was filtered, washed thoroughly with deionized water until neutral, followed by drying at 80 °C for 12 h to obtain RhNiTi-LDHs. Subsequently, the RhNiTi-MMO (RhNiTi-mixed metal oxide) samples were obtained *via* calcining the RhNiTi-LDH precursors at 500 °C in air for 4 h followed by cooling to room temperature. Prior to the catalytic reaction, the RhNiTi-MMO samples were treated in a mixture gas ($H_2/N_2 = 1/9$; flow rate: 50 mL $min^{-1}$) at 400 °C for 2 h to obtain $RhNi/TiO_2$ catalysts. The resulting samples with various Rh content are denoted as $xRhNi/TiO_2$ ($x$ = 0.1, 0.3, 0.5, 0.8, or 1.0), where the $x$ represents the theoretical mass percentage of Rh (e.g., $0.5RhNi/TiO_2$ indicates a 0.5 $wt$.% Rh in this sample). According to the same method described above, the Rh/Ni and $Ni/TiO_2$ catalysts were also prepared without the introduction of Ti and Rh, respectively. In addition, the $Rh/TiO_2$ sample was prepared by traditional impregnation method with 0.5 $wt$.% Rh loading by using anatase as support. The calcination and reduction conditions are consistent with those of $xRhNi/TiO_2$ samples. Inductively coupled plasma atomic emission spectroscopy (ICP-AES) is used to determine the chemical composition of these samples, which is close to the feeding ratio (Supplementary Table 1).

### Catalyst characterizations

The powder XRD (X-ray diffraction) patterns are recorded on a Rigaku XRD-6000 diffractometer using a nickel-filtered Cu $K_\alpha$ radiation source ($\lambda$ = 0.15418 nm) at 40 kV and 30 mA with a scanning rate of 8° $min^{-1}$ and a $2\theta$ angle ranging from 5° to 80°. Crystalline phases are identified by comparison with the reference data from International Center for Diffraction Data (ICDD) files. The chemical composition of various samples is measured by ICP-AES (Shimadzu ICPS-7500). The specific surface area and pore structure parameters of samples are

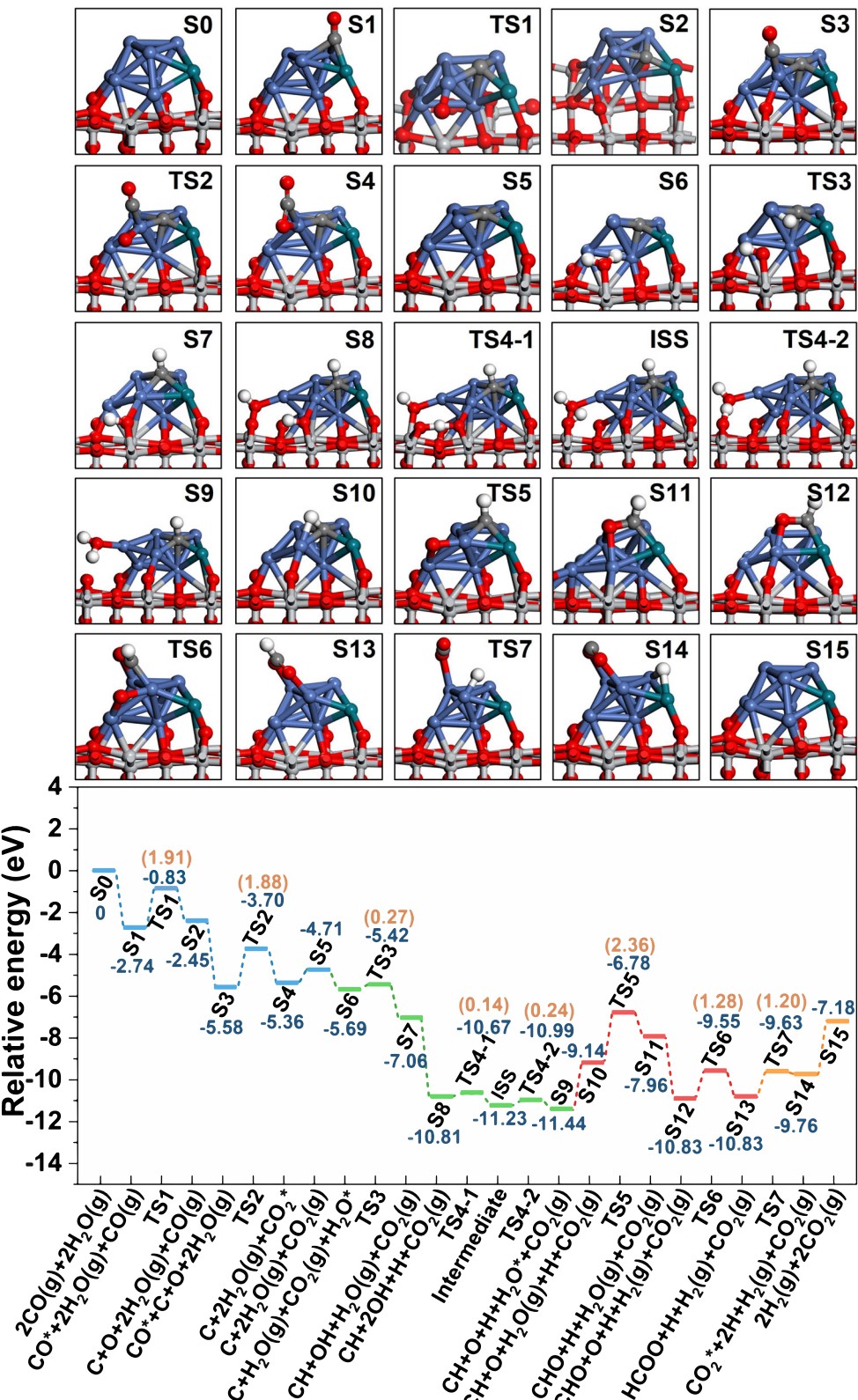

**Fig. 5 | DFT calculation and schematic illustration.** Reaction mechanism of stream reforming of CO on the surface of Rh$_1$Ni$_7$/TiO$_{2-x}$. 'S' denotes a stable sorption state; 'TS' denotes a transition state; 'ISS' denotes an intermedia stable state. (Blue, green, red, and orange dotted lines indicate CO disproportionation, H$_2$O dissociation, formate generation, and CO$_2$ desorption, respectively; blue and orange numbers represent adsorption energy and reaction energy barrier, respectively).

obtained from $N_2$ adsorption and desorption isotherms by using a Quantachrome Auosorb-1C-VP analyzer based on the Brunauer-Emmett-Teller (BET) and Barret-Joyner-Halenda (BJH) models. Sample morphology and structure are characterized by scanning electron microscopy (SEM, Zeiss Supra 55) with applied voltage of 20 kV and transmission electron microscopy (TEM, JEOL JEM-2010) with accelerating voltage of 200 kV. Aberration-corrected high-angle annular dark-field scanning transmission electron microscopy (AC-HAADF-STEM) and element energy-dispersive spectroscopy (EDS) mapping images are conducted on a JEOL JEM-ARM200F equipment. The chemical states of sample surface are investigated by using Thermo VG Escalab 250 X-ray photoelectron spectroscopy (XPS) with Al K$_\alpha$ as a radiation source at 300 W under UHV ($2 \times 10^{-9}$ Torr). The sample after treatment was transferred into sample rod in glove box with $N_2$ atmosphere, and sample charging effects are eliminated by correcting the observed spectra with C 1$s$ binding energy value of 284.8 eV.

Carbon dioxide temperature programmed desorption ($CO_2$-TPD), hydrogen temperature programmed desorption ($H_2$-TPD), and hydrogen temperature programmed reduction ($H_2$-TPR) are carried out on a Micromeritics Chemi-Sorb 2920 instrument equipped with a thermal conductivity detector (TCD). For the $CO_2$-TPD, the sample (0.08 g) was firstly pretreated at 400 °C in a $H_2$ atmosphere for 1 h, followed by purging with He for 0.5 h, and then the temperature was decreased to 50 °C. Subsequently, 5% $CO_2$ was introduced with He as carrier gas until saturation adsorption. Then, pure He was purged, along with the increase of temperature from 50 to 700 °C (rate: 10 °C min$^{-1}$) for the collection of signals. For the $H_2$-TPD, the sample (0.08 g) was firstly pretreated at 400 °C in a $H_2$ atmosphere for 1 h, followed by purging with Ar for 0.5 h, and then the temperature was decreased to 50 °C. Afterwards, a 5% $H_2$ was introduced with Ar as carrier gas until a saturation adsorption. The gas was switched to pure Ar, along with the increase of temperature from 50 to 700 °C (rate: 10 °C min$^{-1}$) for signal collection. For the $H_2$-TPR, the sample (0.1 g) was firstly pretreated at 250 °C in a Ar atmosphere for 1 h, followed by the decrease of temperature to 50 °C. Then, the gas was switched to a 5% $H_2$ with Ar as carrier gas. The temperature was increased from 50 to 600 °C with a rate of 10 °C min$^{-1}$, and meanwhile the $H_2$ consumption signal was recorded.

In situ diffuse reflectance infrared fourier-transform spectra (in situ DRIFTS) are recorded on a VERTEX 70 BRUKER spectrometer equipped with CaF windows and a mercury-cadmium-telluride (MCT) detector, with a resolution of 4 cm$^{-1}$ using 100 scans. The catalyst was filled into an in situ reaction cell and pressed into a flat surface. Firstly, the sample was pretreated at 400 °C in a 10% $H_2$ flow for 1 h followed by purging He for 0.5 h at 400 °C. Prior to the test, the reference baseline was collected; then a switching to $CO/CH_4$ was performed and the diffuse signals were collected at different time points. Furthermore, $H_2O/H_2O + CO_2$ was introduced by a pulse with $N_2$ as carrier gas *via* quantitative loop (1 μL) for signal collection. CO-DRIFT adsorption is carried out at ambiance temperature (20 °C) with CO concentration of 0.5%. For the five-cycle processes of $H_2$ reduction-$O_2$ oxidation with 10%$H_2/N_2$ and air atmosphere, respectively, the reduction and oxidation were performed for 1.0 h at 400 °C. In situ Raman spectra were recorded on a Renishaw in Via-Reflexm equipped with a laser (532 nm). The catalyst was filled into an in situ reaction cell and pretreated in a 10% $H_2$ flow at 400 °C for 1 h followed by purging He at 400 °C for 0.5 h. Afterwards, the Raman signals were collected continually during the adsorption of CO and $H_2O/CO_2$. In situ x-ray absorption fine-structure spectra (in situ XAFS) at Ni K-edge and Ti K-edge are performed at the beamline 1W1B of the Beijing Synchrotron Radiation Facility (BSRF), Institute of High Energy Physics (IHEP), Chinese Academy of Sciences (CAS). The 50 mg of different catalysts were filled into an in situ reaction cell and pretreated in a 10% $H_2$ flow at 400 °C for 1 h followed by purging He at 400 °C for 0.5 h. Afterwards, the signals were collected during the adsorption of CO and $H_2O$. Mass spectral

analysis of CO and $H_2O$ pulsing tests are carried out by Micromeritics Chemi-Sorb 2920 instrument equipped with a mass spectrometry detector (MS).

## Catalytic evaluations for ESR

Catalytic performances of as-synthesized samples toward steam reforming of ethanol (ESR) reaction are studied in a fix-bed reactor with a stainless steel tube (interior diameter: 10 mm) at atmospheric pressure. Prior to the catalytic reaction, 0.15 g of catalyst mixed with quartz sand (40−60 mesh, 1.50 g) was pretreated in a gaseous mixture of $H_2$ and $N_2$ (1:9, v/v; a total gas flow of 50.0 mL min$^{-1}$) at 400 °C for 2 h, and then cooled to reaction temperature (350 °C and 400 °C) in $N_2$ atmosphere. The water and ethanol mixture with steam/carbon (S/C) ratio is 3 was injected into the reaction system by using a HPLC pump at a rate of 0.060 mL min$^{-1}$. The reactants were evaporated in a preheater (170 °C) with a heating belt to avoid product condensation, followed by mixing with nitrogen gas (50.0 mL min$^{-1}$). The temperature of the whole installation was modulated by a K-type thermocouple. When the reaction was carried out at 350 and 400 °C for 1.5 h, the products were analyzed online by using a gas chromatograph (Shimadzu, GC-17A) with both FID and TCD detectors equipped with TDX-01 and HP-PLOT/Q columns, respectively. Ethanol conversion ($X$), product yield ($Y$), and hydrogen selectivity ($S_{H_2}$) are calculated as following equations.

$$X_{EtOH} = \frac{F_{EtOH,in} - F_{EtOH,out}}{F_{EtOH,in}} \times 100\% \qquad (1)$$

$$Y_{H_2} = \frac{F_{H_2}}{6 \times F_{EtOH,in}} \times 100\% \qquad (2)$$

$$Y_{C_i} = \frac{F_{C_i \times j}}{2 \times F_{EtOH,in}} \times 100\% \qquad (3)$$

$$S_{H_2} = \frac{F_{H_2}}{F_{H_2} + 2 \times F_{CH_4} + 2 \times F_{CH_3CHO}} \times 100\% \qquad (4)$$

$F_{EtOH,in/out}$ is the molar flow rate of ethanol at the inlet/outlet of the reactor, respectively. $F_{H_2}$ and $F_{C_i}$ denote the molar flow rate of $H_2$ and C-containing product at the reactor outlet, respectively, where the $j$ indicates the number of carbon atoms in the latter. The molar flow of acetaldehyde, ethylene, and methane is determined based on FID results. The molar flow of other gas products ($H_2$, CO and $CO_2$) in the effluent are measured by TCD results, which are calculated based on Eqs. (5) and (6):

$$V_j = V \times A_j \times \frac{y_j^{std}}{A_j^{std}} \qquad (5)$$

$$PV = nRT \qquad (6)$$

where $T$, $P$, R, $n$, and $V$ are the room temperature (K), pressure (Pa), molar gas constant (8.314 J mol$^{-1}$ K$^{-1}$), molar number, and total volumetric flow of the gas outlet, respectively. $A_j$ is the peak area of component $j$ obtained from TCD signal. $A_j^{std}$ and $y_j^{std}$ are the peak area and the molar fraction of component $j$ in the standard gas mixture, respectively.

## Reaction dynamics analysis

For the measurement of reaction rate of ethanol and acetaldehyde, CO, and $CH_4$ steam reforming reaction, the quartz sand ($SiO_2$, 50−300 mg) and the catalyst sample (5−30 mg) are sieved separately

(40−60 mesh) and then physically mixed together before installed into the reactor tube. Reaction conditions for kinetics studies over 0.5RhNi/TiO₂, Ni/TiO₂, Rh/Ni, and Rh/TiO₂ catalysts are as follows: liquid feed of ethanol and acetaldehyde at 0.040 mL min⁻¹, liquid feed of H₂O at 0.032 mL min⁻¹, gas flow rate of CO at 50 mL min⁻¹ and CH₄ at 10.0 mL min⁻¹, CO₂/H₂ at 10.0 mL min⁻¹, N₂ carrier at 50 mL min⁻¹, reaction temperature: 240−400 °C, time on stream: 0.5 h. For the determination of reaction order of ethanol dehydrogenation and acetaldehyde decomposition, the initial partial pressure of ethanol and acetaldehyde are 7−39 kPa and 4−15 kPa, respectively. For the determination of the reaction order of $CH_4$ steam reforming, the initial partial pressure of $CH_4$ is 4−12 kPa. For the determination of the reaction order of CO steam reforming, the initial partial pressure of $H_2O$, CO, $CO_2$, and $H_2$ is tuned within 20−40 kPa, 4−17 kPa, 3−13 kPa, and 4−23 kPa, respectively.

## Computational details

Density functional theory (DFT) calculations are performed in Vienna ab initio simulation package (VASP) with the generalized gradient approximation (GGA) using the Perdew-Burke-Ernzerhof (PBE) functional[46]. The projected augmented wave (PAW) potentials are used to describe the ionic cores, and valence electrons are also considered using a plane wave basis set with a kinetic energy cutoff of 400 eV[47]. Geometry optimizations are performed with the force convergence smaller than 0.05 eV Å⁻¹, where the same convergency is applied for the location of transition states by the constrained optimizations. The original bulk structure is optimized before the construction of surfaces with the Monkhorst-Pack k-point of 3 × 3 × 1. The TiO₂(110) surface with 24 Ti and 48 O atoms is applied with half of the atoms at the bottom fixed in all the calculations. A Ni₈ cluster with 8 Ni atoms is placed on the TiO₂(110) surface to describe the interface of Ni/TiO₂. According to the STEM results, one Ni atom is then replaced by one Rh atom on Ni₈ surface to build the RhNi/TiO₂ interface structure. A Monkhorst-Pack k-point 3 × 3 × 1 is applied for all the calculations on surfaces. In addition, the effect from the Hubbard U corrections is considered beyond the accuracy of DFT calculations of GGA, where $U$ value (employed as U-J) of 3.5 is applied for Ti, Ni, and Rh.

Transition state (TS) searches are performed at the same theoretical level with the CI-NEB method. All the models are the most stable structure obtained through optimization and screening. The formation energy ($FE_{Ov}$) is used in analyzing oxygen vacancy formation (O$_v$), defined as

$$FE_{Ov} = E_{vo-slab} + E_{o-gas} - E_{slab} \qquad (7)$$

where $E_{vo\text{-}slab}$, $E_{o\text{-}gas}$, and $E_{slab}$ are the total energies for the oxygen vacancy slab, the oxygen atom in the gas phase, and the clean surface, respectively.

The adsorption energy ($E_{ads}$) is calculated as

$$E_{ads} = E_{total} - \left( E_{slab} + E_g \right) \qquad (8)$$

where $E_{total}$ is the total energy after adsorption; $E_{slab}$ is the energy of the clean slab before adsorption, and $E_g$ is the energy of the free adsorbate in the gas phase.

The energy barrier ($E_a$) is obtained from the electronic energy difference between the transition state ($E_{TS}$) and its corresponding initial state ($E_{IS}$), which is calculated by

$$E_a = E_{TS} - E_{IS} \qquad (9)$$

In this work, a Ni₇Rh₁/TiO₂₋ₓ model is applied for calculation, in which Ni₇Rh₁ cluster is supported on the TiO₂(110) surface with oxygen vacancy. Before the calculations, the model of Ni₇Rh₁/TiO₂₋ₓ is optimized. The lattice parameters of TiO₂ support are: $a = b = 3.79$ Å,

$c = 9.56$ Å, $\alpha = \beta = \gamma = 90°$ (body-centered tetragonal) with a p(2 × 2) supercell.

## Data availability

The primary data that support the plots within this paper and other finding of this study are available from the corresponding author on reasonable request. Source data are provided with this paper.

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

## Acknowledgements

This work was supported by the National Natural Science Foundation of China (22172006, 22102006, 22002007, 22272007, U19B6002 and 22288102), the National Key R&D Program of China (2021YFC2103500), the Beijing Natural Science Foundation (2212012), and the Fundamental Research Funds for the Central Universities (XK2022-12). The authors are thankful for the support of the BSRF (Beijing Synchrotron Radiation Facility) during the XAFS measurements at the beamline of 1W1B and 1W2B.

## Author contributions

H.M. performed the catalyst preparation and characterizations. H.M. and J.Z. performed the catalytic evaluations. H.M. and Y.Y. prepared the draft manuscript. T.S., P.Y., and H.Y. performed the DFT theoretical calculations. B.Z., Y.N., L.Z., W.L., L.W., and Z.R. participated in the catalyst structure investigations. Y.Y., J.Z., F.-S.X., M.W., and X.D. designed the study, analyzed the data, and revised the manuscript.

## Competing interests

The authors declare no competing interests.
