## [Peer Review File · Nature Communications]

A strong bimetal-support interaction in ethanol steam reformingREVIEWER COMMENTS

Reviewer #1 (Remarks to the Author):

0.5RhNi/TiO₂ catalyst was synthesized and illustrated to be effective ESR. Various analysis including DFT calculations were performed to understand the performance of the synthesized catalysts. Most of the work was carried out carefully and the results seem to be significant. However, I will only recommend its publication after resolving the following concerns.

1) hydrogen production of ESR is dependent on the ration of S/C. on p.5, it states as 3 but in figures, such as Fig.1, it is 6. Which one is correct? A direct comparison in Table 2 of supporting information cannot be made directly if the current work is 6.

2) authors claimed that the rate-determining step is only the formation of formate intermediate. This assumption is not supported by the results of Fig.6 of supporting information. C-C bond cleavage is as rate-determining as the formation of formate intermediate. Furthermore, the difference between (b) and (e) in Fig.1 also does not support the conclusion drawn by authors. The recent work on the competition between dehydrogenation and C-C bond cleavage on Ir [ChemPhysChem (2022)e202200132 by Wu, et al] may provide some insight on the selection of the intermediate(s) to choose for C-C bond cleavage in DFT calculations to confirm.

3) The chosen reactions in the DFT calculations are not critical ones to support the conclusion of experimental observations. The DFT calculations show that $\text{CO} \rightarrow \text{C} + \text{O}$ cannot take place as the barrier is too high. Therefore, $\text{C} + \text{H}_2\text{O} \rightarrow \text{CH} + \text{OH}$ is not feasible as no C species. Water dissociation will most likely occur via a normal path, as shown in J. Phys. Chem. 113(2009)7269; J. Phys. Chem. C 124(2020)26953, $\text{H}_2\text{O} \rightarrow \text{H} + \text{OH}$.

4) Eq(2) and Eq(4) did not include water in the calculation but water provides half of H atoms to form H₂ in a perfect ESR.

Reviewer #2 (Remarks to the Author):

1. Page 5, section Catalytic performance and kinetic analysis toward ESR: The comparison of catalysts at total conversion is meaningless and does not have any justification. Such results only mean that ethanol is exhausted by the reaction but they do not allow us to explain how the catalysts work. Such results can be a consequence of an excess of contact time (i.e. of catalyst mass) for the test conditions selected. In such a situation, it is unclear if the all active sites of the catalyst sample in the catalytic bed participate in reactions or some active sites cannot participate in it because ethanol is exhausted before reaching the catalyst bed. It is also possible that activity decay is not observed because, if the catalyst amount is in excess, decay is compensated by a higher amount of catalyst taking part in the reaction (without still reaching the total amount of the bed). The catalytic tests presented in Figure 1b should be done once more for higher space velocity.

2. Page 5, section Catalytic performance and kinetic analysis toward ESR: What was the time on stream for the catalytic tests presented in Figure 1 b? A piece of information about it must be written in the Experimental part.

3. Page 5, section Catalytic performance and kinetic analysis toward ESR: Based on the caption of Supplementary Figure 6, it could be supposed that the time on stream for catalytic tests performed at temperatures of 350 and 400 °C equals 1.5 hours. Why this time is so short? If the Authors run the catalytic tests longer, there is possible that the deactivation of some catalysts could be observed.

4. Page 8, section Catalytic performance and kinetic analysis toward ESR: 'Furthermore, we performed kinetic tests of CO and CH₄ steam reforming reaction to simulate the reaction rate of CO and CH_x intermediates during ESR reaction (Fig. 1e).' The Authors indicate that performed catalytic tests. As it is well known these kinds of tests should be carried out at low ethanol conversion. Thus, the Authors should indicate in Figure in Supplementary materials that the conversion of ethanol during these tests was lower than 10%.

5. Page 22: Taking into account the amount of research carried out for the catalysts studied in the ESR process, the conclusions are rather short and poor and must be rewritten.

Response to Reviewers

Reviewer #1

Remarks to the Author:

0.5RhNi/TiO₂ catalyst was synthesized and illustrated to be effective ESR. Various analysis including DFT calculations were performed to understand the performance of the synthesized catalysts. Most of the work was carried out carefully and the results seem to be significant. However, I will only recommend its publication after resolving the following concerns.

(1) Hydrogen production of ESR is dependent on the ration of S/C. on p.5, it states as 3 but in figures, such as Fig.1, it is 6. Which one is correct? A direct comparison in Table 2 of supporting information cannot be made directly if the current work is 6.

Author reply: Thank you very much for this comment. On Page 5, the value of 3 is the steam/carbon ratio (S/C), where one ethanol molecule provides two carbon atom and thus the ratio is 3. However, in the caption of Fig. 1, the value of 6 denotes the molar ratio of water and ethanol for the inlet gas. Both values are correct with different expressions. Thus, in Table 2 of supporting information, the S/C ratio is 3. According to the suggestions above, in order to avoid misunderstanding, we adopted a unified form of steam/carbon ratio (S/C) of 3 in the revised manuscript.

● **Page 5, Line 18: rephrase:** “The ESR reaction was performed in a fixed-bed reactor with steam/carbon (S/C) ratio of 3.”

● **Page 7, the caption in Fig. 1b and c: rephrase:** “reaction conditions: catalyst (0.15 g) + SiO₂ (1.50 g); liquid feed of S/C = 3, at 0.060 mL min⁻¹; N₂ carrier at 50.0 mL min⁻¹; reaction temperature: 400 °C; time on stream: 1.5 h.”

● **Page 8, the caption in Fig. 1h and i: rephrase:** “MS signals from the *operando* pulse experiment of ethanol and water (S/C = 3) over 0.5RhNi/TiO₂ catalyst at 400 °C.”

(2) authors claimed that the rate-determining step is only the formation of formate intermediate. This assumption is not supported by the results of Fig. 6 of supporting information. C-C bond cleavage is as rate-determining as the formation of formate intermediate. Furthermore, the difference between (b) and (e) in Fig.1 also does not support the conclusion drawn by authors. The recent work on the competition between dehydrogenation and C-C bond cleavage on Ir [ChemPhysChem (2022)202200132 by Wu, et al] may provide some insight on the selection of the intermediate(s) to choose for C-C bond cleavage in DFT calculations to confirm.

Author reply: Thank you for this comment. To clearly identify the rate-determining step, we measured the apparent activation energy and reaction order of ethanol dehydrogenation, acetaldehyde decomposition, steam reforming of CO (or CH₄) based on kinetics studies in the revised manuscript (Supplementary Figure 14). In addition, according to the report mentioned above, we also calculated the reaction energy barrier for ethanol dehydrogenation (C–H bond cleavage) and acetaldehyde decomposition (C–C bond cleavage) through DFT calculations (Supplementary Fig. 53). Corresponding discussions have been added in the revised manuscript.

• **Page 8, Line 17: rephrase:** “Furthermore, we performed kinetic tests of ethanol dehydrogenation, acetaldehyde decomposition, steam reforming of CO (or CH₄) to simulate the reactivity of C–H bond cleavage, C–C bond cleavage, CO or CH_x transformation during ESR reaction. As shown in Supplementary Fig. 14a, the activation energy of elementary reaction gives the following sequence: ethanol dehydrogenation (44.54 kJ/mol) < acetaldehyde decomposition (50.45 kJ/mol) < CO steam reforming (89.46 kJ/mol) < CH₄ steam reforming (101.26 kJ/mol), which indicates that the cleavage of C–H and C–C bonds is facile whilst the transformation of CO and CH_x is involved in the rate-determining step. This result is further demonstrated through a more significant concentration-dependence of reaction order for CO and CH₄ relative to ethanol and acetaldehyde: ethanol (0.46) < acetaldehyde (0.62) < CO (1.18) < CH₄ (1.23) (Supplementary Fig. 14b).”

• **Page 20, Line 9: rephrase:** “In addition, we also calculated the reaction energy barriers for ethanol dehydrogenation and acetaldehyde decomposition through DFT calculations

(Supplementary Fig. 53). The formation of formate (intermediate of CO and CH_x transformation) shows an energy barrier of 2.36 eV, much larger than that of ethanol dehydrogenation (C–H bond cleavage: 1.03 eV) and acetaldehyde decomposition (C–C bond cleavage: 1.17 eV). This verifies that the transformation of CO and CH_x is involved in the rate-determining step, in accordance with the experimental results.”

- Supplementary Figure 14 and 53 have been supplemented in the revised SI.

Supplementary Figure 14. **a** Apparent activation energy and **b** reaction order tests for ethanol dehydrogenation, acetaldehyde decomposition, steam reforming of CO (or CH₄) within kinetic range (conversion less than 10%). Reaction conditions: catalyst (5–30 mg) + SiO₂ (50–300 mg); liquid feed of ethanol (or acetaldehyde) at 0.04 mL min⁻¹, CO at 50 mL min⁻¹, CH₄ at 10 mL min⁻¹, N₂ carrier at 50 mL min⁻¹, reaction temperature: 240–360 °C, time on stream: 0.5 h.

Supplementary Figure 53. Calculated potential energy diagram and corresponding geometric structures for ethanol dehydrogenation and acetaldehyde decomposition (IS, TS and FS represent the initial state, transition state and final state, respectively).

(3) The chosen reactions in the DFT calculations are not critical ones to support the conclusion of experimental observations. The DFT calculations show that $\text{CO} \rightarrow \text{C} + \text{O}$ cannot take place as the barrier is too high. Therefore, $\text{C} + \text{H}_2\text{O} \rightarrow \text{CH} + \text{OH}$ is not feasible as no C species. Water dissociation will most likely occur via a normal path, as shown in *J. Phys. Chem.* 113(2009)7269; *J. Phys. Chem. C* 124(2020)26953, $\text{H}_2\text{O} \rightarrow \text{H} + \text{OH}$.

Author reply: Thank you for this comment. The computational scheme is mainly derived from the experimental results. For the *in situ* DRIFT and *in situ* Raman spectra (Fig. 4a–c and Supplementary Figure 34 and 35), both the CO_2 and C signals are observed after the introduction of CO at 400 °C, which is consistent with the CO-disproportionation route ($2\text{CO} \rightarrow \text{CO}_2 + \text{C}$). According to the comments above, we recalculated the energy barrier of CO-disproportionation, including CO direct dissociation pathway ($\text{CO} \rightarrow \text{C} + \text{O}$; $\text{CO} + \text{O} \rightarrow \text{CO}_2$) and double CO association dissociation pathway ($2\text{CO} \rightarrow \text{C} + \text{CO}_2$). In addition, the water

dissociation ($\text{H}_2\text{O} \rightarrow \text{H} + \text{OH}$) at the interface oxygen vacancy was also calculated. Corresponding discussions have been added in Supplementary Note 22 in the revised SI.

• **Supplementary Information, Page 82, Line 5: rephrase:** “For the CO disproportionation process, two reaction pathways have been reported including CO direct dissociation pathway ($\text{CO} \rightarrow \text{C} + \text{O}$; $\text{CO} + \text{O} \rightarrow \text{CO}_2$) and double CO association dissociation pathway ($2\text{CO} \rightarrow \text{C} + \text{CO}_2$). In the first case (Supplementary Fig. 54: R1), one CO molecule undergoes adsorption at the hollow site near two Ni atoms and one Rh atom with a dissociation energy barrier of 1.91 eV; and the resulting active oxygen is embedded in the oxygen vacancy of TiO_{2-x} . Then, another one CO molecule reacts with interfacial O species to generate CO_2 with an energy barrier of 1.88 eV (Supplementary Fig. 54: R2), followed by CO_2 desorption from the catalyst surface to complete CO disproportionation reaction (Supplementary Fig. 54: R3). In contrast, for the double CO association dissociation, two CO molecules experience activation adsorption at adjacent Ni sites to form CO-CO structure, followed by dissociation to produce CO_2 and C species with a relatively high energy barrier of 2.84 eV (Supplementary Fig. 55). Therefore, CO direct dissociation is the thermodynamically favorable pathway.”

• **Supplementary Information, Page 80, Line 17: rephrase:** “The water dissociation at the interface oxygen vacancy was also calculated. Two dissociation paths have been reported including hydroxyl self-disproportionation and water continuous dissociation path. In the former case (Supplementary Fig. 56), H_2O molecule firstly undergoes adsorption dissociation at the interface oxygen vacancy to generate hydrogen atom and hydroxyl species ($\text{H}_2\text{O} \rightarrow \text{H} + \text{OH}$). Regardless of the presence of C species on the catalyst surface (Supplementary Fig. 56: R1 and Supplementary Fig. 57: R1), the water dissociation displays similar energy barriers (0.27 and 0.28 eV), indicating that the surface C species has no effect on water activation dissociation due to the different adsorption sites for C and H_2O . Afterwards, two hydroxyls experience disproportionation to produce reactive oxygen species ($2\text{OH} \rightarrow \text{H}_2\text{O} + \text{O}$) with an energy barrier of 0.24 eV (Supplementary Figure 56: R3). This process shows a much lower energy barrier compared with the hydroxyl continuous dissociation path ($\text{OH} \rightarrow \text{O} + \text{H}$) with an energy barrier of 0.94 eV (Supplementary Figure 57: R2). Thus, the former is the dominant pathway for the generation of reactive oxygen species.”

- Supplementary Figure 54–57 have been supplemented in the revised SI.

Supplementary Figure 54. Calculated potential energy diagram and corresponding geometric structures for CO direct dissociation pathway (IS, TS and FS represent the initial state, transition state and final state, respectively).

Supplementary Figure 55. Calculated potential energy diagram and corresponding geometric structures for double CO association dissociation pathway (IS, TS and FS represent the initial state, transition state and final state, respectively).

Supplementary Figure 56. Calculated potential energy diagram and corresponding geometric structures for H₂O dissociation to active oxygen followed by hydroxyl self-disproportionation pathway (IS, TS and FS represent the initial state, transition state and final state, respectively).

Supplementary Figure 57. Calculated potential energy diagram and corresponding geometric structures for H_2O continuous dissociation pathway (IS, TS and FS represent the initial state, transition state and final state, respectively).

(4) Eq(2) and Eq(4) did not include water in the calculation but water provides half of H atoms to form H_2 in a perfect ESR.

Author reply: Thank you for this comment. Actually, this issue (half of hydrogen provided by water) has been considered and included in Eq(2): $Y_{\text{H}_2} = \frac{F_{\text{H}_2}}{6 \times F_{\text{EtOH,in}}} \times 100\%$. Based on the chemical reaction equation: $\text{C}_2\text{H}_6\text{O} + 3\text{H}_2\text{O} \rightarrow 2\text{CO}_2 + 6\text{H}_2$, the consumption of one ethanol molecule would theoretically produce six hydrogen molecules. Thus, the coefficient for H_2 is 1 and the coefficient for ethanol is 6 in Eq(2). The similar method has been widely used to calculate hydrogen yield in ESR (*Appl. Catal., B: Environ.* 293 (2021) 120178; *Chem. Eng. J.* 326 (2017) 956; *Appl. Catal., B: Environ.* 145 (2014) 73). In the case of Eq(4): $S_{\text{H}_2} = \frac{F_{\text{H}_2}}{F_{\text{H}_2} + 2 \times F_{\text{CH}_4} + 2 \times F_{\text{CH}_3\text{CHO}}} \times 100\%$, where the H_2 selectivity is calculated based on the proportion of hydrogen atoms in H_2 and other products (CH_4 and CH_3CHO). Because water is not a product but a reactant, the H atoms resulting from water dissociation is also included in F_{H_2} . The similar calculated method has been employed in previous studies (*Chem. Eng. J.* 379 (2020) 122299; *J. Catal.* 416 (2022) 240; *J. Catal.* 393 (2021) 159).

Reviewer #2

Remarks to the Author:

(1) Page 5, section Catalytic performance and kinetic analysis toward ESR: The

comparison of catalysts at total conversion is meaningless and does not have any justification. Such results only mean that ethanol is exhausted by the reaction but they do not allow us to explain how the catalysts work. Such results can be a consequence of an excess of contact time (i.e. of catalyst mass) for the test conditions selected. In such a situation, it is unclear if the all active sites of the catalyst sample in the catalytic bed participate in reactions or some active sites cannot participate in it because ethanol is exhausted before reaching the catalytic bed. It is also possible that activity decay is not observed because, if the catalyst amount is in excess, decay is compensated by a higher amount of catalyst taking part in the reaction (without still reaching the total amount of the bed). The catalytic tests presented in Figure 1b should be done once more for higher space velocity.

Author reply: Thank you for this comment. We re-tested the catalytic performance of Ni/TiO₂, 0.5RhNi/TiO₂, Rh/Ni and Rh/TiO₂ samples at a higher weight hourly space velocity (WHSV: 28 h⁻¹) and gas hourly space velocity (GHSV: 16700 h⁻¹) by reducing the catalyst dosage from 0.15 g to 0.05 g, and other reaction conditions remained unchanged (Supplementary Fig. 8). Corresponding discussion has been added in the revised manuscript.

● **Page 6, Line 7: rephrase:** “Furthermore, we also tested the ESR reaction at a higher WHSV (21 h⁻¹) and GHSV (16700 h⁻¹) at 400 °C (ethanol conversion less than 70%), and the 0.5RhNi/TiO₂ catalyst still displayed the optimal hydrogen yield and relatively low CH₄ and CO yields (Supplementary Fig. 8).”

● Supplementary Figure 8 has been supplemented in the revised SI.

Supplementary Figure 8. Ethanol conversion and product yield over various catalysts. ESR reaction conditions: catalyst (0.05 g) + SiO₂ (0.50 g); S/C = 3 at 0.060 mL min⁻¹; N₂ carrier at 50.0 mL min⁻¹; reaction temperature: 400 °C; time on stream: 1.5 h.

(2) Page 5, section Catalytic performance and kinetic analysis toward ESR: What was the time on stream for the catalytic tests presented in Figure 1 b? A piece of information about it must be written in the Experimental part.

Author reply: Thank you for this comment. The time on stream is 1.5 h for the catalytic tests in Figure 1b. According to this comment, the related information has been added in the caption of Fig. 1 and the experimental section in the revised manuscript.

● **Page 7, the caption in Fig. 1b: rephrase:** “Ethanol conversion, products yield and H₂ production rate over various catalysts (reaction conditions: catalyst (0.15 g) + SiO₂ (1.50 g); S/C = 3, at 0.060 mL min⁻¹; N₂ carrier at 50.0 mL min⁻¹; reaction temperature: 400 °C; time on stream: 1.5 h).”

● **Page 27, Line 20: rephrase:** “When the reaction was carried out at 350 and 400 °C for 1.5 h, the products were analyzed online by using a gas chromatograph (Shimadzu, GC-17A) with both FID and TCD detectors equipped with TDX-01 and HP-PLOT/Q columns, respectively.”

(3) Page 5, section Catalytic performance and kinetic analysis toward ESR: Based on the caption of Supplementary Figure 6, it could be supposed that the time on stream for catalytic tests performed at temperatures of 350 and 400 °C equals 1.5 hours. Why this time is so short? If the Authors run the catalytic tests longer, there is possible that the deactivation of some catalysts could be observed.

Author reply: Thank you for this comment. According to this comment, we carried out time on stream (TOS) tests for 0.5RhNi/TiO₂, Ni/TiO₂, Rh/Ni and Rh/TiO₂ at 400 °C, respectively, and the results were shown in Supplementary Fig. 9 in the revised SI. The related discussion has been added in the revised manuscript.

● **Page 6, Line 12: rephrase:** “In addition, the time on stream (TOS) tests for Ni/TiO₂, Rh/Ni,

Rh/TiO₂ and 0.5RhNi/TiO₂ were carried out at 400 °C, respectively. After reaction for 40 h, the ethanol conversion over Ni/TiO₂, Rh/Ni, and Rh/TiO₂ decreases from 100% to 91.33%, 58.54% and 74.16%, respectively (Supplementary Fig. 9). In contrast, both the ethanol conversion and hydrogen yield in the presence of 0.5RhNi/TiO₂ catalyst remain stable within 300 h (Fig. 1d).”

• Supplementary Figure 9 has been supplemented in the revised SI.

Supplementary Figure 9. Time on stream (TOS) tests for Ni/TiO₂, 0.5RhNi/TiO₂, Rh/Ni and Rh/TiO₂ at 400 °C, respectively.

(4) Page 8, section Catalytic performance and kinetic analysis toward ESR: ‘Furthermore, we performed kinetic tests of CO and CH₄ steam reforming reaction to simulate the reaction rate of CO and CH_x intermediates during ESR reaction (Fig. 1e).’ The authors indicate that performed catalytic tests. As it is well known these kinds of tests should be carried out at low ethanol conversion. Thus, the Authors should indicate in Figure in Supplementary materials that the conversion of ethanol during these tests was lower than 10%.

Author reply: Thank you for this comment. For the kinetic tests, we did proceed at a low conversion of ethanol, CO and CH₄ (less than 10%). The relevant description has been added in the caption of Figure 1 and Supplementary Figure 14.

• **Page 7, the caption of Fig. 1: rephrase:** “Reaction rate for CO/CH₄ steam reforming reaction within kinetic range (CO or CH₄ conversion less than 10%). Reaction conditions: catalyst (5

mg) + SiO₂ (50 mg), liquid feed of H₂O at 0.032 mL min⁻¹, CO at 50.0 mL min⁻¹, CH₄ at 10 mL min⁻¹, CO₂/H₂ at 10.0 mL min⁻¹, N₂ carrier at 50 mL min⁻¹, reaction temperature: 400 °C, time on stream: 0.5 h.”

• **Supplementary information, Page 20, the caption of Supplementary Figure 14: rephrase:** “Apparent activation energy and reaction order for ethanol dehydrogenation, acetaldehyde decomposition, steam reforming of CO (or CH₄) within kinetic range (conversion less than 10%). Reaction conditions: catalyst (5–30 mg) + SiO₂ (50–300 mg), liquid feed of ethanol (or acetaldehyde) at 0.04 mL min⁻¹, CO at 50 mL min⁻¹, CH₄ at 10 at mL min⁻¹, N₂ carrier at 50 mL min⁻¹, reaction temperature: 240–360 °C, time on stream: 0.5 h.”

(5) Page 22: Taking into account the amount of research carried out for the catalysts studied in the ESR process, the conclusions are rather short and poor and must be rewritten.

Author reply: Thank you for this comment. According to this suggestion, we have improved the conclusion in the revised manuscript.

• **Page 23, Line 4: rephrase:** “In summary, we report a RhNi/TiO₂ catalytic system with well-defined SBMSI towards ESR reaction. The obtained 0.5RhNi/TiO₂ catalyst gives an exceptional hydrogen production (H₂ yield: 61.7%) and catalysis stability (300 h) at a relatively low temperature (400 °C). The microscopic fine-structure of RhNi/TiO₂ was studied by STEM, CO-DRIFT and XAFS, in which the RhNi bimetallic nanoparticle with a reversible TiO₂ coating exhibited a multiple electron transfer pathway at the interfacial active sites (Rh-Ni^{δ-}-O_v-Ti³⁺). A comprehensive investigation including *in situ* spectroscopic characterizations, *operando* pulse experiments, kinetics studies and DFT calculations substantiates that the ESR reaction in the presence of 0.5RhNi/TiO₂ catalyst follows a CO/CH_x-mediated reforming process rather than the acetate path. This bimetal-support interface (Rh-Ni^{δ-}-O_v-Ti³⁺) plays a decisive role in steam reforming of CO and CH_x originating from ethanol dissociation, which is involved in the rate-determining step of ESR reaction. The modulated geometric and electronic structure of interfacial active sites resulting from SBMSI reduce the binding ability of species with COO⁻ structure (CO₂, carbonate or formate). This facilitates the generation and

transformation of formate intermediate from CO/CH_x reforming processes, which ensures a prominent hydrogen production rate and catalytic stability. The well-defined SBMSI demonstrated in this work can be extended to other structure-sensitive reactions involving multiple reaction substrates.”

REVIEWER COMMENTS

Reviewer #1 (Remarks to the Author):

Authors made efforts in the revision to address the concerns by the referees. Although improvements are made, I will only recommend its publication after resolving the following concerns.

- 1) Reaction mechanism of ESR is extremely complicated. Authors claimed in the response that they calculated the reaction energy barrier for ethanol dehydrogenation according to the report mentioned (i.e. recommended reference). There is an issue: the authors only choose to calculate the barrier for $\text{CH}_3\text{CH}_2\text{OH} \rightarrow \text{CH}_3\text{CH}_2\text{O}$, which is not the case shown in the reference. At least some justification is needed on why they choose only this dehydrogenation when three possible pathways are feasible and needed to study before concluding it is the one. Furthermore, in recent bimetallic studies, the relative energetics of reaction pathways may change due to alloying (is especially true in authors' work) and brings some extra challenge to the computational studies of ESR mechanism, see Wu, et al, J. Phys. Chem. 126(2022)21650, and therefore caution should be made when using limited DFT studies.
- 2) Experimental measurements shown in supplementary Figure 14 are great. When they are correlated wrongly with the elementary reactions, conclusion can mislead readers. Therefore, a link between the observations and the DFT results needs to be established carefully so that the results of the work are impactful.

Response to Reviewers

Reviewer #1

Remarks to the Author:

Authors made efforts in the revision to address the concerns by the referees. Although improvements are made, I will only recommend its publication after resolving the following concerns.

(1) Reaction mechanism of ESR is extremely complicated. Authors claimed in the response that they calculated the reaction energy barrier for ethanol dehydrogenation according to the report mentioned (i.e. recommended reference). There is an issue: the authors only choose to calculate the barrier for $\text{CH}_3\text{CH}_2\text{OH} \rightarrow \text{CH}_3\text{CH}_2\text{O}$, which is not the case shown in the reference. At least some justification is needed on why they choose only this dehydrogenation when three possible pathways are feasible and needed to study before concluding it is the one. Furthermore, in recent bimetallic studies, the relative energetics of reaction pathways may change due to alloying (is especially true in authors' work) and brings some extra challenge to the computational studies of ESR mechanism, see Wu, et al, *J. Phys. Chem.* 126(2022)21650, and therefore caution should be made when using limited DFT studies.

Author reply: Thank you very much for this comment. Based on the product distribution and *operando* characterizations (DRIFTS spectra and pulse experiment), important reaction intermediates (e.g., $\text{CH}_3\text{CH}_2\text{O}$, CH_3CHO , CH_3 and CO) were captured, which verified the ESR reaction pathway over the RhNi/TiO_{2-x} catalyst. Ethanol undergoes dehydrogenation to acetaldehyde, followed by acetaldehyde decomposition to CO and CH_x . Subsequently, the resulting CO and CH_x as key intermediates react with H_2O to produce CO_2 and H_2 . Thus, we speculate experimentally that ethanol dehydrogenation involves the cleavage of O–H (in

hydroxy) and C–H (in methylene rather than methyl) bonds to produce CH_3CO^* . According to this comment, we supplemented the ethanol dehydrogenation steps on $\text{Rh}_1\text{Ni}_7/\text{TiO}_{2-x}$ and $\text{Ni}_8/\text{TiO}_{2-x}$ systems through DFT calculations. In addition, comparative studies on ESR reaction mechanism were also conducted with mono-metal system ($\text{Ni}_8/\text{TiO}_{2-x}$), and the corresponding discussions have been added in the revised manuscript and Supplementary Information.

● **Page 20, Line 9: rephrase:** “For the ESR reaction mechanism, the reaction energy barriers for ethanol dehydrogenation and acetaldehyde decomposition were studied through DFT calculations. Based on the calculation results (Supplementary Figs. 53–62, Supplementary Table 8 and Supplementary Note 22), the optimal ethanol dehydrogenation path in $\text{Ni}_8/\text{TiO}_{2-x}$ and $\text{Rh}_1\text{Ni}_7/\text{TiO}_{2-x}$ systems follows $\text{CH}_3\text{CH}_2\text{OH} \rightarrow \text{CH}_3\text{CH}_2\text{O}^* \rightarrow \text{CH}_3\text{CHO}^* \rightarrow \text{CH}_3\text{CO}^*$, and then CH_3CO^* undergoes C–C bond breaking to produce CH_3^* and CO, which is consistent with the product distribution and *operando* characterizations (DRIFTS spectra and pulse experiment). Compared with $\text{Ni}_8/\text{TiO}_{2-x}$, the energy barriers for ethanol dehydrogenation and C–C bond cleavage decrease from 1.65 and 1.31 eV to 1.03 and 1.17 eV on $\text{Rh}_1\text{Ni}_7/\text{TiO}_{2-x}$, respectively.”

● **Page 21, Line 14: rephrase:** “In contrast, the formate formation from C/CH fragment shows an energy barrier of 2.36 and 2.79 eV on $\text{Rh}_1\text{Ni}_7/\text{TiO}_{2-x}$ and $\text{Ni}_8/\text{TiO}_{2-x}$ catalysts, respectively, much larger than that of ethanol dehydrogenation (1.03 and 1.65 eV) and acetaldehyde decomposition (1.17 and 1.31 eV), indicating that the transformation of CO and CH_x is the crucial step, in accordance with the experimental results. Especially, the lower reaction energy barriers on $\text{Rh}_1\text{Ni}_7/\text{TiO}_{2-x}$ relative to $\text{Ni}_8/\text{TiO}_{2-x}$ verify that the ESR reaction is boosted at the bimetal-support interface sites, in well agreement with the catalytic evaluations.”

● **Supplementary Information, Page 82, Supplementary Note 22: rephrase:** “As shown in Supplementary Figs. 53–58 and Supplementary Table 8, various ethanol dehydrogenation routes were calculated on $\text{Rh}_1\text{Ni}_7/\text{TiO}_{2-x}$ and $\text{Ni}_8/\text{TiO}_{2-x}$ samples, respectively, and the surface metal sites act as the active center in favor of ethanol activation adsorption. For Route 1

($\text{CH}_3\text{CH}_2\text{OH} \rightarrow \text{CH}_3\text{CH}_2\text{O}^* \rightarrow \text{CH}_3\text{CHO}^* \rightarrow \text{CH}_3\text{CO}^*$), Route 2 ($\text{CH}_3\text{CH}_2\text{OH} \rightarrow \text{CH}_3\text{CHOH}^* \rightarrow \text{CH}_3\text{CHO}^* \rightarrow \text{CH}_3\text{CO}^*$) and Route 3 ($\text{CH}_3\text{CH}_2\text{OH} \rightarrow \text{CH}_3\text{CHOH}^* \rightarrow \text{CH}_3\text{COH}^* \rightarrow \text{CH}_3\text{CO}^*$), the reaction energy barriers over $\text{Rh}_1\text{Ni}_7/\text{TiO}_{2-x}$ and $\text{Ni}_8/\text{TiO}_{2-x}$ are 1.03, 4.06, 4.06 eV and 1.65, 2.63, 3.18 eV, respectively. For the methyl dehydrogenation in ethanol ($\text{CH}_3\text{CH}_2\text{OH} \rightarrow \text{CH}_2\text{CH}_2\text{OH}$), the reaction energy barriers are 2.03 and 2.18 eV over $\text{Rh}_1\text{Ni}_7/\text{TiO}_{2-x}$ and $\text{Ni}_8/\text{TiO}_{2-x}$ catalysts, respectively (Supplementary Figs. 59 and 60). Remarkably, the lowest reaction energy barrier for Route 1 indicates that the successive dehydrogenation of ethanol molecule (*via* hydroxy, methylene and methine groups) is the optimal path on both catalysts, which is consistent with the experimental results. Subsequently, the CH_3CO species undergoes C–C bond cleavage to generate CO and CH_3 with energy barriers of 1.17 and 1.31 eV on $\text{Rh}_1\text{Ni}_7/\text{TiO}_{2-x}$ and $\text{Ni}_8/\text{TiO}_{2-x}$ catalysts, respectively (Supplementary Figs. 61 and 62). Compared with $\text{Ni}_8/\text{TiO}_{2-x}$, the largely decreased reaction energy barriers for ethanol dehydrogenation and acetaldehyde decomposition on $\text{Rh}_1\text{Ni}_7/\text{TiO}_{2-x}$ demonstrate the advantages of bimetallic synergistic effect towards ethanol conversion.”

• **Supplementary Figs. 53–62, 69–73 and Supplementary Table 8 have been supplemented in the revised Supplementary Information.**

Supplementary Figure 53. Calculated potential energy diagram and corresponding geometric structures for successive dehydrogenation of ethanol molecule *via* hydroxy, methylene and methine groups (Route 1) on $\text{Rh}_1\text{Ni}_7/\text{TiO}_{2-x}$ (IS, TS and FS represent the initial state, transition state and final state, respectively; black and red numbers denote adsorption energy and reaction energy barrier, respectively).

Supplementary Figure 54. Calculated potential energy diagram and corresponding geometric structures for successive dehydrogenation of ethanol molecule *via* methylene, hydroxy and methine groups (Route 2) on $\text{Rh}_1\text{Ni}_7/\text{TiO}_{2-x}$ (IS, TS and FS represent the initial state, transition state and final state, respectively; black and red numbers denote adsorption energy and reaction energy barrier, respectively).

Supplementary Figure 55. Calculated potential energy diagram and corresponding geometric structures for successive dehydrogenation of ethanol molecule *via* methylene, methine and hydroxy groups (Route 3) on $\text{Rh}_1\text{Ni}_7/\text{TiO}_{2-x}$ (IS, TS and FS represent the initial state, transition state and final state, respectively; black and red numbers denote adsorption energy and reaction energy barrier, respectively).

Supplementary Figure 56. Calculated potential energy diagram and corresponding geometric structures for successive dehydrogenation of ethanol molecule *via* hydroxy, methylene and methine groups (Route 1) on $\text{Ni}_8/\text{TiO}_{2-x}$ (IS, TS and FS represent the initial state, transition state and final state, respectively; black and red numbers denote adsorption energy and reaction energy barrier, respectively).

Supplementary Figure 57. Calculated potential energy diagram and corresponding geometric structures for successive dehydrogenation of ethanol molecule *via* methylene, hydroxy and methine groups (Route 2) on $\text{Ni}_8/\text{TiO}_{2-x}$ (IS, TS and FS represent the initial state, transition state and final state, respectively; black and red numbers denote adsorption energy and reaction energy barrier, respectively).

Supplementary Figure 58. Calculated potential energy diagram and corresponding geometric structures for successive dehydrogenation of ethanol molecule *via* methylene, methine and hydroxy groups (Route 3) on Ni₈/TiO_{2-x} (IS, TS and FS represent the initial state, transition state and final state, respectively; black and red numbers denote adsorption energy and reaction energy barrier, respectively).

Supplementary Figure 59. Calculated potential energy diagram and corresponding geometric structures for methyl dehydrogenation in ethanol on $\text{Rh}_1\text{Ni}_7/\text{TiO}_{2-x}$ (IS, TS and FS represent the initial state, transition state and final state, respectively; black and red numbers denote adsorption energy and reaction energy barrier, respectively).

Supplementary Figure 60. Calculated potential energy diagram and corresponding geometric structures for methyl dehydrogenation in ethanol on $\text{Ni}_8/\text{TiO}_{2-x}$ (IS, TS and FS represent the initial state, transition state and final state, respectively; black and red numbers denote adsorption energy and reaction energy barrier, respectively).

Supplementary Figure 61. Calculated potential energy diagram and corresponding geometric structures for C–C bond cleavage in CH₃CO* over Rh₁Ni₇/TiO_{2-x} (IS, TS and FS represent the initial state, transition state and final state, respectively; black and red numbers denote adsorption energy and reaction energy barrier, respectively).

Supplementary Figure 62. Calculated potential energy diagram and corresponding geometric structures for C–C bond cleavage in CH₃CO* over Ni₈/TiO_{2-x} (IS, TS and FS represent the initial state, transition state and final state, respectively; black and red numbers denote adsorption energy and reaction energy barrier, respectively).

Supplementary Figure 69. Calculated potential energy diagram and corresponding geometric structures for CO disproportionation reaction on $\text{Ni}_8/\text{TiO}_{2-x}$ (IS, TS and FS represent the initial state, transition state and final state, respectively).

Supplementary Figure 70. Calculated potential energy diagram and corresponding geometric structures for H₂O dissociation to active oxygen on Ni₈/TiO_{2-x} (IS, TS and FS represent the initial state, transition state and final state, respectively).

Supplementary Figure 71. Calculated potential energy diagram and corresponding geometric structures for the generation of carboxylate from HCO* and active oxygen species, followed by a subsequent transition to formate intermediate on Ni₈/TiO_{2-x} (IS, TS and FS represent the initial state, transition state and final state, respectively).

Supplementary Figure 72. Calculated potential energy diagram and corresponding geometric structures for the generation of carboxylate from COH^* and active oxygen species, followed by a subsequent transition to formate intermediate on Ni_8/TiO_{2-x} (IS, TS and FS represent the initial state, transition state and final state, respectively).

Supplementary Figure 73. Reaction mechanism of stream reforming of CO on the surface of Ni₈/TiO_{2-x} (S, TS and ISS represent a stable sorption state, a transition state and an intermediate stable state, respectively). Blue, green, red and orange dotted lines denote CO disproportionation, H₂O dissociation, formate generation and CO₂ desorption, respectively; blue and orange numbers represent adsorption energy and reaction energy barrier, respectively).

Supplementary Table 8. Calculated activation energy (E_a) and reaction energy (ΔE) of various elementary reactions in ESR over $\text{Rh}_1\text{Ni}_7/\text{TiO}_{2-x}$ and $\text{Ni}_8/\text{TiO}_{2-x}$ systems

Elementary Reaction		$\text{Rh}_1\text{Ni}_7/\text{TiO}_{2-x}$		$\text{Ni}_8/\text{TiO}_{2-x}$	
		E_a (eV)	ΔE (eV)	E_a (eV)	ΔE (eV)
CH₃CH₂OH dehydrogenation via Route 1	$\text{CH}_3\text{CH}_2\text{OH}^* \rightarrow \text{CH}_3\text{CH}_2\text{O}^*$	1.03	-1.00	0.59	-0.45
	$\text{CH}_3\text{CH}_2\text{O}^* \rightarrow \text{CH}_3\text{CHO}^*$	1.01	-0.01	0.86	0.07
	$\text{CH}_3\text{CHO}^* \rightarrow \text{CH}_3\text{CO}^*$	0.72	0.08	1.65	1.46
CH₃CH₂OH dehydrogenation via Route 2	$\text{CH}_3\text{CH}_2\text{OH}^* \rightarrow \text{CH}_3\text{CHOH}^*$	4.06	1.29	2.63	1.68
	$\text{CH}_3\text{CHOH}^* \rightarrow \text{CH}_3\text{CHO}^*$	3.35	-0.23	0.97	-1.58
	$\text{CH}_3\text{CHO}^* \rightarrow \text{CH}_3\text{CO}^*$	0.83	-0.45	1.64	1.21
CH₃CH₂OH dehydrogenation via Route 3	$\text{CH}_3\text{CH}_2\text{OH}^* \rightarrow \text{CH}_3\text{CHOH}^*$	4.06	1.29	2.63	1.68
	$\text{CH}_3\text{CHOH}^* \rightarrow \text{CH}_3\text{COH}^*$	0.53	-0.32	3.18	0.74
	$\text{CH}_3\text{COH}^* \rightarrow \text{CH}_3\text{CO}^*$	0.78	-0.79	0.63	-1.16
CH₃ dehydrogenation in CH₃CH₂OH	$\text{CH}_3\text{CH}_2\text{OH}^* \rightarrow \text{CH}_2\text{CH}_2\text{OH}^*$	2.03	1.53	2.18	0.80
C-C bond cleavage	$\text{CH}_3\text{CO}^* \rightarrow \text{CH}_3^* + \text{CO}^*$	1.17	-0.45	1.31	-0.69
CH₃ successive dehydrogenation	$\text{CH}_3^* \rightarrow \text{CH}_2^* + \text{H}$	0.64	0.38	0.54	0.37
	$\text{CH}_2^* \rightarrow \text{CH}^* + \text{H}$	0.87	-0.53	1.17	0.80
	$\text{CH}^* \rightarrow \text{C}^* + \text{H}$	2.79	-0.36	2.94	0.18
CO disproportionation	$\text{CO}^* \rightarrow \text{C}^* + \text{O}^*$	1.91	0.29	2.17	0.62
	$\text{C}^* + \text{O}^* + \text{CO}^* \rightarrow \text{C}^* + \text{CO}_2^*$	1.88	0.22	2.30	0.62
CO₂ desorption	$\text{CO}_2^* \rightarrow \text{CO}_2(\text{g})$	-	0.65	-	0.93
H₂O dissociation to active oxygen	$\text{C}^* + \text{H}_2\text{O}^* \rightarrow \text{CH}^* + \text{OH}^*$	0.27	-1.37	0.30	-0.68
	$\text{OH}^* \rightarrow \text{O}^* + \text{H}$	0.94	-1.45	1.50	-0.20
	$\text{CH}^* + 2\text{OH}^* + \text{H} \rightarrow \text{CH}^* + \text{O}^* + \text{H} + \text{H}_2\text{O}^*$	0.24	-0.21	0.65	-0.24
H₂O desorption	$\text{H}_2\text{O}^* \rightarrow \text{H}_2\text{O}(\text{g})$	-	2.30	-	3.54
Formate generation via HCO*	$\text{CH}^* + \text{O}^* \rightarrow \text{HCO}^*$	2.36	1.18	1.63	0.84
	$\text{HCO}^* + \text{O}^* \rightarrow \text{HCOO}^*$	1.28	0	2.79	0.52
	$\text{HCOO}^* \rightarrow \text{CO}_2^* + \text{H}$	1.20	1.07	2.71	1.29
Formate generation via COH*	$\text{C}^* + \text{OH}^* \rightarrow \text{COH}^*$	3.28	1.22	4.04	1.01
	$\text{COH}^* + \text{O} \rightarrow \text{COOH}^*$	0.91	-0.17	3.59	0.75
	$\text{COOH}^* \rightarrow \text{HCOO}^*$	4.16	0.26	-	-
CO₂ desorption	$\text{CO}_2^* \rightarrow \text{CO}_2(\text{g})$	-	2.58	-	2.67

2) Experimental measurements shown in supplementary Figure 14 are great. When they are correlated wrongly with the elementary reactions, conclusion can mislead readers. Therefore, a link between the observations and the DFT results needs to be established carefully so that the results of the work are impactful.

Author reply: Thank you for this comment. According to this suggestion, we have improved the correlations between experimental observations and DFT calculation results in the revised manuscript.

• **Page 8, Line 17: rephrase:** “Furthermore, we performed kinetic tests on ethanol dehydrogenation, acetaldehyde decomposition, steam reforming of CO (or CH₄) to study the C–H bond cleavage, C–C bond cleavage, CO or CH_x transformation during ESR reaction. As shown in Supplementary Fig. 14a, the apparent activation energy of these reaction processes gives the following sequence: ethanol dehydrogenation (44.54 kJ mol⁻¹) < acetaldehyde decomposition (50.45 kJ mol⁻¹) < CO steam reforming (89.46 kJ mol⁻¹) < CH₄ steam reforming (101.26 kJ mol⁻¹), which indicates that the cleavage of C–H and C–C bonds in ethanol is facile whilst the transformation of intermediates (CO and CH_x) is rather difficult. This result is further demonstrated through a more significantly concentration-dependent reaction order for CO and CH₄ in comparison with ethanol and acetaldehyde: ethanol (0.46) < acetaldehyde (0.62) < CO (1.18) < CH₄ (1.23) (Supplementary Fig. 14b).”

REVIEWERS' COMMENTS

Reviewer #1 (Remarks to the Author):

the authors performed DFT calculations and made corresponding revision. I ahve no other concerns.

Response to Reviewers

Reviewer #1 (Remarks to the Author):

The authors performed DFT calculations and made corresponding revision. I have no other concerns.

Author reply: Thank you for this comment.